# Aroma Patterns Characterization of Braised Pork Obtained from a Novel Ingredient by Sensory-Guided Analysis and Gas-Chromatography-Olfactometry

**DOI:** 10.3390/foods8030087

**Published:** 2019-03-02

**Authors:** Shiqing Song, Li Fan, Xiaodong Xu, Rui Xu, Qian Jia, Tao Feng

**Affiliations:** School of perfume and aroma technology, Shanghai Institute of Technology, No.100 Hai Quan Road, Shanghai 201418, China; sshqingg@163.com (S.S.); 18721984426@163.com (L.F.); 18255003001@163.com (X.X.); 18621053365@163.com (R.X.); 176071204@mail.sit.edu.cn (Q.J.)

**Keywords:** braised pork, braised sauce, aroma compounds, aroma extract dilution analysis, odor activity value

## Abstract

Two types of braised pork were prepared from self-made braised sauce added to Maillard reaction intermediate (MRI) and white granulated sugar, respectively. Descriptive sensory analysis and gas chromatography-mass spectrometry (GC-MS) were conducted to investigate their differences in sensory and aroma compounds. The results showed that the effect of self-made braised sauce in braised pork was comparable to white granulated sugar. One-hundred-and-nine volatile flavor compounds were identified by GC-MS using headspace-solid phase microextraction (HS-SPME) and simultaneous distillation and extraction (SDE). Thirty-six odor active compounds with retention indexes ranging from 935–2465 were identified by aroma extract dilution analysis (AEDA). Additionally, their odor activity values (OAV) were calculated. It was found that 17 aroma compounds showed an OAV greater than 1. Among them, pentanal (almond, pungent), nonanal (fat, green), (E, E)-2,4-decadienal (fat, roast), phenylacetaldehyde (hawthorn, honey, sweet), dodecanal (lily, fat, citrus) and linalool (floral, lavender) reached the highest OAV values (>200), indicating a significant contribution to the aroma of two types of braised pork. These results indicated that the self-made braised sauce added with MRI could be used for cooking braised pork with good sensory characteristics.

## 1. Introduction

Braised pork is a traditional Chinese cuisine made from pig meat, which is quite popular among ordinary people due to its mellow taste and fatty but not greasy characteristics. There are several kinds of braised pork, such as Maoist braised pork (peppery taste), Dongpo braised pork (soft and refreshing), and Shanghai braised pork (sweet salty palatability). However, in order to reach a perfect color and flavor, too much sugar need to be added added during the cooking of braised pork. This may not only result in the decline of the nutritional value of the food itself but also excessive sugar consumption has been identified as the major cause of excessive caloric intake and is the main dietary determinant of obesity or other civilization diseases.

Recent researches have focused on the new sweeteners or seasonings as substitutions for sugar in food formulas [1,2]. However, these sweeteners cannot produce intense aroma and color compared with the white granulated sugar. It is well known that Maillard reaction plays an important role in the generation of aroma and taste-active compounds of processed food [3]. Complete Maillard reaction products (MRPs) are usually used for savory flavoring [4]. They have been favored by consumers over the years and are found to be intriguing by researchers because of their rich fragrance. However, since most of the flavor components in MRPs are volatile, it is difficult to maintain the stability of flavored end products for application, especially during the heating treatment of cooking. The loss of aroma and fragrance makes MRPs limited in the application. During the initial stage of Maillard reaction, neither aromas nor melanoids are produced but significant non-volatile aroma precursors-Maillard reaction intermediate (MRI) are formed [5]. When heated, MRI undergoes dehydration and fission as the Maillard reaction progresses and generates colorless reductones and heterocyclic compounds [6]. Hence, the braised sauce with MRI as a flavoring additive cannot exhibit its flavor during primary processing or storage at room temperature but generates desirable flavor and color during cooking.

However, there is lack of deep research on braised sauce with added MRI. Thus, the MRI from xylose-L-cysteine was prepared in an aqueous phase and its applications in braised pork were tested by gas chromatography-olfactometry (GC-O) with aroma extraction dilution analysis (AEDA) and odor activity values (OAV). Xylose, an aldopentose, is one of the reaction precursors. It is also mainly used as an ideal sweetener because of its physical characteristics, such as being a non-digestible sugar and having zero calories. L-cysteine (cys) was an important precursor for the formation of sulphur compounds and has been extensively used in the manufacturing of reaction flavors. Many researches have been reported to detect aroma-active compounds using GC-O [7,8,9,10,11,12]. In addition, in order to obtain the more comprehensive analysis of flavor compounds, solid phase microextraction (SPME) and simultaneous distillation and extraction (SDE) pre-treatment methods were compared.

During this study, two types of braised pork were prepared, cooked with self-made braised sauce added with MRI (A) and white granulated sugar (B), respectively. The objectives of the present study were (1) to compare the sensory characteristics and aroma attributes of the two kinds of braised pork by descriptive sensory analysis, (2) to analyze and compare the volatile compounds by SPME/SDE-GC-MS, (3) to screen the characteristic aroma-active compounds as determined by AEDA and OAV, and (4) to assess through the above analysis and whether the self-made braised sauce containing MRI was recommended for new cooking methods for braised pork.

## 2. Materials and Methods 

### 2.1. Chemicals

3-methylbutanal, pentanal, α-pinene, hexanal, β-pinene, δ-3-carene, limonene, 1,8-cineole, acetoin, 2-methyltetrahydrofuran-3-one, nonanal, furfural, 2-acetylfuran, benzaldehyde, linalool, 1-terpinen-4-ol, (E)-2-decenal, phenylacetaldehyde, furfuryl alcohol, (E)-cinnamaldehyde, neral, dodecanal, 3-phenylpropanal, anethole, (E,E)-2,4-decadienal, 2-methoxybenzaldehyde, 2-acetylpyrrole, 4-methoxybenzaldehyde, (*Z*)-cinnamaldehyde, phenol, ethyl cinnamate, γ-undecalactone, cinnamyl alcohol, and coumarin. 

### 2.2. Materials to Food Sample

Streaky pork (Shuanghui, China), edible soya oil (Golden arowana, China) and all other food materials were purchased from the local Tesco supermarket. 1,2-dichlorobenzene (internal standard) and C6-C30 n-alkane series (concentration of 1000 mg/L in in hexane) were purchased from Sigma-Aldrich Chemical Co. (St. Louis, MO, USA). The five-spice powder, braised soy sauce and pork paste were purchased from Tesco supermarket. The refined lard was purchased from Anhui Muyang Oil and Fats Co., Ltd. (Anhui, China).

### 2.3. Preparation of Self-Made Braised Sauce 

Preparation of MRI: The MRI was prepared according to the Cui et al., method with some modification [13]. Xylose (4.5 g) and L-cysteine (0.54 g) (molar ratio about 10:1.5) were dissolved in water (50 g). The solution was transferred into 100-mL screw-sealed tubes. The pH was adjusted to 7.0 with either 1 mol/L HCl or 6 mol/L NaOH, the tubes were tightly capped and then heated in an oil bath with magnetic stirring (150 rpm) at 100 °C for 80 min. After the reaction, the tubes were immediately cooled in ice water for further use. The preparation was carried out in triplicate.

Preparation of braised sauce: The optimum formula of braised sauce was determined by a single factor test. The final recipe was as follows: MRI solution 82.80%, xylose 16.56%, methionine 0.34%, monascus color 0.07%, caramel pigment 0.10%, ethyl maltol 0.03%, and xanthan gum 0.10%. All ingredients were blended, stirred for 5 minutes at 400 r/min and then dispersed by a high-shear dispersion homogenizer at 10000 r/min for 3 minutes to a homogeneous solution. The preparation was carried out in triplicate.

### 2.4. Preparation of Braised Pork

The preparation was as follows: Step 1: The power of induction cooker was set to 1300 W, 10 g oil was put into the frying pan and 10 g Chinese onion; 10 g ginger and 10 g garlic were stirred into it for about 1 minute. Then 500 g streaky pork (3 centimeter square and 4 centimeter thick pieces) was transferred to the pan and quickly fried until the meat turned white. At that point, 50 g yellow wine, 12.5 g dark soy sauce and 80 g self-made braised sauce were spooned over the meat and stirred for about 4 minutes. Step 2: The heat was turned down to 800 W, and 5 g cinnamon, 5 g star anise, and 1000 g warm water were added to the above material and left to simmer very gently for 30 minutes until thickened. As the final step, the material was seasoned with 0.5 g salt and stirred rapidly under 1600 W power until the gravy turned thick, then it was dished up and served immediately. The braised pork sample was named as A.

The preparation of the traditional braised pork was the same as A but the 80-g self-made braised sauce was substituted with 20 g white granulated sugar and it was named as B.

### 2.5. Sensory Evaluation

The sensory analysis of the two types of braised pork was performed in a sensory laboratory set in accordance with ISO 8589 (2007). A ten-member panel (five females and five males, aged from 22 to 35 years) was selected to participate in the discriminating and descriptive evaluations. They were trained for 2 weeks to familiarize them with the characteristic of braised pork to be evaluated. In addition, panelists had stated and discussed characteristic sample aromas through three preliminary sessions until all of them had agreed on a consensus vocabulary to describe samples. Then, seven aroma attributes including meat flavor, spicy flavor, greasy, sauce flavor, braised flavor, caramel flavor, and overall flavor were used for the descriptive analysis. To ensure that the panelists clarified these sensory attributes, the reference materials were set as follows: Streaky pork (200 g) wrapped in aluminum foil and baked at 150 °C for 1 h for a “meat” note, a kind of commercially available five-spice powder dissolved in hot water for a “spicy” note, refined lard (100 g) for a “greasy” note, a kind of commercially available braise soy sauce for a “sauce” note, a kind of commercially available pork paste for a “braised” note, and white granulated sugar (100 g) boiled on a small fire to obtain a caramel flavor. The panelist was instructed to score on a scale from 0 (not detectable) to 9 (strongly detectable) with 9-point category scale properties in comparison with the standard flavor models [14]. An evaluation was done in triplicate (on separate days) in isolated booths under controlled light (artificial daylight) and room temperature (21 ± 1 °C) [15].

### 2.6. SPME 

Divinylbenzene/carboxen/polydimethylsiloxane (DVB/CAR/PDMS) fiber (57 mm length; 50/30 µm film thickness) was adopted. A total of 6 g braised pork (fat 3.6 g, lean meat 2.4 g) was added into a 15 mL amber vial closed by a PTFE/silicone septum. Before the extraction process, a time of 15 min at 60 °C was given for subsequent headspace equilibration; the SPME fiber was exposed in the headspace, placed for 30 min, and then thermally desorbed in the GC-MS injector port. 

### 2.7. SDE

An SDE was performed in a Likense Nickerson apparatus (SDE apparatus) using dichloromethane as the extraction solvent. A total of 200 g of braised pork (fat 120 g, lean meat 80 g) was immersed in a flask with 250 mL of distilled water, and 80 mL of dichloromethane was put in another flask. Both flasks were heated up to their boiling points. Once the two flasks started to reflux, the distillation–extraction was continued for 4 h to allow the volatile components to collect in dichloromethane. After cooling to ambient temperature for 10 min, the dichloromethane extract was dried over anhydrous Na_2_SO_4_ which was maintained at −18 °C overnight. The extract was filtered through a folded filter paper with favorable hydrophilic properties. The filtrate was then collected and concentrated to 2 mL in a rotary evaporator and then to exactly 0.5 mL under gentle nitrogen flow at room temperature. Concentrated extract was kept at −24 °C for GC-MS-O and GC-MS analysis.

### 2.8. GC-MS Analysis

The analysis of volatile compounds obtained by SPME and SDE was performed on a GC-MS using an Agilent 5975C mass selective detector coupled with an Agilent 7890A GC (Agilent, Santa Clara, CA, USA), equipped with HP-Wax capillary column (60 m × 0.25 mm inner diameter, 0.25 µm film thickness). Helium was used as a carrier gas and the flow rate was 1 mL/min. For the SDE, 1 µL extract was injected in splitless mode at 250 °C; for SPME analysis, desorption was also in splitless mode at 250 °C for 5 min. GC oven temperature was programmed at 40 °C (held for 6 min), increased to 100 °C at 3 °C/min, and finally increased to 230 °C (SPME held for 10 min, SDE held for 20 min) at 5 °C/min. The MS conditions were as follows: The transfer line temperature was 280 °C; the ion source temperature was 230 °C; and ionization energy 70 eV, and mass range, 20 to 350 a.m.u.

All experiments were carried out in triplicate and the results were reported as mean values.

### 2.9. SDE-GC-O Analysis

GC separations were carried out on an Agilent 6890N instrument equipped with a split-splitless injector, an FID (flame ionization detector) and a Gerstel ODP (olfactory detection port). Approximately, 1 μL of each SDE extract was injected in splitless mode into a capillary column (HP-Wax column, 60 m × 0.25 mm inner diameter, 0.25 µm film thickness). The oven conditions, injector and detector temperatures were the same as those given above for GC-MS. Nitrogen was used as the carrier gas at a flow rate of 1 mL/min. The split ratio was 10:1. The eluate was split to 1:1 at the end of the capillary into the FID detector and the ODP device.

### 2.10. Aroma Extraction Dilution Analysis (AEDA)

The SDE extract was stepwise (3-fold, 1:3, 1:9, 1:27, 1:81, 1:243…) diluted with dichloromethane until the sniffer could not detect any significant odor. Each flavor was thus assigned a Flavor Dilution (FD) factor representing the last dilution in which the odor was still detectable. GC-O was performed by three experienced panelists. The aroma description and sniffing time of each compound were determined by at least two panelists.

### 2.11. Qualitative and Quantitative Analysis of the Volatile Compounds

The volatile compounds were identified either by comparing the retention indices (RIs) and mass fragmented patterns with those of reference compounds, or by matching with mass spectrums in WILEY and NIST database and reported RI values. The RI of volatile compounds was calculated based on a C6–C30 n-alkane series that were injected under the same chromatographic conditions as for test samples. The quantities of the odor active compounds were estimated by an external standard method.

### 2.12. Calibration of Standard Curves

Odor-active compounds identified by SDE-GC-O were quantitated by constructing standard curves. The mixed standard solutions were prepared through dissolving standard compounds in dichloromethane. Six levels of concentration were prepared for the calibration, and standards were analyzed in triplicate. In order to make the quantitative results more accurate, 16 aldehydes and 18 non-aldehydes were divided into two groups. Ten mg of each standard compound and 0.1 g of the internal standard solution (1,2-dichlorobenzene, 10 μg/g) were introduced to 5 g dichloromethane. The standard stock solution was then diluted with dichloromethane for six levels (1:10, 1:20, 1:30, 1:40, 1:50, and 1:60) for the calibration. The calibration equation for each compound was carried out by plotting the response ratio of standard compounds and 1,2-dichlorobenzene against their concentration ratio. 

These solutions were analyzed using HP-Wax column by GC–MS as described in Section 2.8, except that mass spectrometry was conducted in the single ion monitoring (SIM) mode. By the calibration equation, the concentration of each odor-active compound was calculated. The final results were the average of three replicates. The limits of detection (LOD) were estimated as the concentration of a standard whose signal-to-noise ratio was 3. The limits of quantitation (LOQ) were estimated as the concentration of a standard whose signal-to-noise ratio was 10.

### 2.13. Odor Aroma Value Determination

OAV (odor activity value) was determined by dividing the concentration by its odor threshold. OAV = Wi/OTi, where Wi is the concentration (μg/g braised pork) of compound i, and OTi is its odor detection threshold concentration in water (μg/g) which was found in the literature. The compound with OAV ≥ 1 was identified as the key aroma component in braised pork.

### 2.14. Statistical Analysis

The data were analyzed by one-way analysis of variance ANOVA using the statistical software SAS 9.1.3 (SAS Institute Inc., USA). An interaction analysis (3-way ANOVA) was performed by the software PanelCheck V1.3.2. Significant effects were performed using Tukey’s least significant difference (LSD) test at the level of 0.05 (*p* ≤ 0.05). 

## 3. Results

### 3.1. Sensory Evaluation of the Braised Pork Samples

The sensory differences between different braised porks were evaluated. Significant differences among two samples for all attributes (Figure 1) indicated that the tested samples had different aroma intensities except for spicy flavor, probably because they had the same addition of spicy materials. An interaction analysis shows that there were no significant panelists, and the replication effect (*p* < 0.05) was found for all attributes. In addition, no significant interactions between sample × panelist and panelist × replication were found, showing that all the panelists were reproducible in the triplicate tests for each attribute. However, a significant interaction between sample and replication was observed for the “Overall flavor” (*p* < 0.05) attributes, indicating that the intensities of the attribute in the samples were not rated similarly when they were replicated.

In Chinese traditional food culture, braised flavor, sauce flavor, and caramel flavor are part of the typical fragrance of braised pork in various categories. An extremely significant influence (*p* < 0.001) was found for braised flavor, which might be partly affected by the different Maillard reaction effect of sucrose and xylose. Also, the overall flavor (*p* < 0.05), meat flavor (*p* < 0.05), greasy (*p* < 0.01), and sauce flavor (*p* < 0.01) of sample A were significantly higher than B. Instead, the score of caramel flavor for sample A was lower than sample B probably due to the white granulated sugar added. However, the GC-MS results show that many compounds such as furfural, 5-methylfurfural and 2,3-dihydro-3,5-dihydroxy-6-methyl-4H-pyran-4-one which might be generated via the caramelization of sugars were found in sample A in larger amounts than in sample B. This contradiction is normal because of the complexity of the food matrix and flavor generation. Nevertheless, the sensory panel agreed that the effect of self-made braised sauce in braised pork was as good as white granulated sugar and the lower sweetness of sample A was more in line with people’s needs for healthy eating.

### 3.2. Volatile Flavor Compounds Identified from the Braised Pork

The flavor substances were separated first in order to identify them. The methods for flavor substances separation include solvent extraction, distillation extraction, supercritical fluid extraction, and headspace capture. Generally, in the extraction process, the most widely used were the simultaneous distillation extraction (SDE) and solid phase micro-extraction (SPME). The SDE method has a good effect on the extraction of low volatility and high boiling components, such as esters and hydrocarbons with complex molecular structures. The SPME method adsorbs the sample directly, the sample processing time is short, the temperature is low, the steps are short, and the volatile components detected do not change.

The results from the two extraction methods SDE and SPME are summarized according to the functional groups in Table 1 and Table 2. A total of 109 aroma compounds were identified, of these, aldehydes were the most predominant in number (24 compounds), followed by alcohols, oxygen-containing heterocycles, acids, and ketones. Seventy-seven compounds were detected using the SPME method, while 68 compounds were detected in SDE extracts and 36 compounds co-existed in both methods. Methanethiol was the most abundant aroma substance in SPME, whereas anethole was the most abundant in SDE. The difference should be due to the fact that methanethiol is a very volatile compound while anethole is a moderately volatile compound and it is proved that volatile flavor profiles differ according to the extraction method. Thus, these two methods can be deemed as a complement for each other during the volatility analysis of braised pork. 

Comparing the volatiles extracted by the two techniques, it could be revealed that SDE extracted a higher number of aldehydes, ketones, acids and phenols than the SPME, while the SPME extracted more hydrocarbons, sulphur, and nitrogen compounds. It was worth noting that sulphur-containing compounds were not detected in SDE and only one nitrogenous compound (2-acetylpyrrole) was identified. This result might be attributed to the evaporation step during the SDE process that might lead to more complex side reactions and the loss of some volatile compounds.

From the semi quantitation results, the most abundant class of compounds from SPME was the sulphur compounds in both samples, followed by aldehydes. A different situation was observed in SDE; the most abundant class was the aldehydes, followed by ethers. Furthermore, aldehydes considered to be formed by the mono-unsaturated fatty acids and Strecker oxidative degradation of amino acids, which made a great contribution to the flavor of braised pork due to their low odor threshold [19,20,21]. Nine kinds of aldehyde compounds were commonly identified in these two methods including 3-methylbutanal, pentanal, hexanal, heptanal, benzaldehyde, phenylacetaldehyde, (E,E)-2,4-decadienal, 4-methoxybenzaldehyde and (*Z*)-cinnamaldehyde. Moreover, SPME detected more low-molecular-weight aldehydes, such as hexanal, heptanal, (E)-2-heptenal, and nonanal. Meanwhile, SDE detected more long chain and methoxy aldehydes such as (Z)-13-octadecenal and 2-methoxybenzaldehyde.

Comparing the SPME results of the two braised pork samples with each other, sample A tended to be more dominant in the kinds of volatiles. Moreover, a higher number and number of oxygen-containing heterocycles and nitrogenous compounds were identified in sample A compared to sample B. In contrast, the content of sulphur compounds in sample B was slightly higher. Some sulfur compounds are allyl mercaptan derivatives which occur via the degradation of the precursors in garlic material [22]. Some other compounds were the result of a Maillard reaction between sugar and amino acid contributing to the meaty aroma, roasted flavor, and caramel-like odor. Among these volatiles, methanthiol was the most abundant and might have originated from methionine degradation [23]. It was highly volatile and easily extracted by the fiber. However, due to its strong volatility, it could easily be lost during extraction and this may explain why it was not detected in the SDE method. 3-methylbutanal was also detected in relatively high quantities and could be generated from Strecker degradation of leucine [24,25]. It was found that 3-methylbutanal was only detected in sample B. Besides, methional with sauce flavor is a meat-flavoring compound, and was detected only in sample A, which might have contributed to the sauce flavor of sample A [26].

Furan ring compounds can be obtained by 1,2-enolization and cyclization of amino acid-Amadori compound [27,28]. Furthermore, Table 1 showed that thiophene, 2-methyl-3-sulfanylfuran, and furfuryl mercaptan were only detected in sample A; the reason for this was probably due to the addition of MRI.

The results of SDE analysis show that sample B exhibited a more dominant number of volatile components than sample A. The discrepancy between the two samples mainly comes from the content difference of aldehydes and lactone. It was found that sample B had a higher peak area percentage of benzaldehyde, phenylacetaldehyde, 3-phenylpropanal, and (*Z*)-cinnamaldehyde. However, only one lactone γ-undecalactone was detected in sample B. 

### 3.3. Identification of the Odor-Active Compounds in Braised Pork

In the above analysis, the results showed that the compounds obtained by the two extraction methods (SPME and SDE) had complementary effects. However, from a quantitative perspective, only the SDE extraction method was used for odor-active compounds analysis. Table 3 summarizes the aroma-active compounds detected in the extracts of two types of braised pork by AEDA. A total of 36 aroma-active compounds were observed by AEDA, including 16 aldehydes, 4 hydrocarbons, 3 alcohols, 1 ketone, 2 esters, 2 ethers, 1 phenol, 1 nitrogenous compound, 4 oxygen-containing heterocycles, and 2 unknown compounds. Thirty compounds were detected in sample A and 28 in sample B. The FD profiles of the aroma-active volatile compounds are shown in Figure 2. The RI of the larger FD factor (FD ≥ 27) aroma compounds fell within the range of RI 1135–2053, and these compounds were selected as potent compounds in the SDE of two types of braised pork extracts by AEDA. Furfuryl alcohol (sauce-flavor), anethole (anise, slight sweet), and unknown compound 1 were proved to be the most powerful aroma-active ones with the same highest FD factor of 243. Based on the sniff description of above aroma-active compounds, it was concluded that the main aroma of braised pork was meaty, caramel, sauce flavor, and spicy.

For sample A, furfuryl alcohol (sauce-flavor) was the most intense aroma-active compound owning the highest aroma intensity (FD = 243). Pentanal (almond, pungent), limonene (fruity, orange), furfural (bread, almond, sweet), and anethole (anise, light sweet aroma) had the higher FD factor (FD = 81) which also suggested them to be the key contributors to the overall flavor. Besides, there were 12 kinds of substances with FD = 27, such as α-pinene, δ-3-carene, 1,8-cineole, 2-acetylfuran, ethyl cinnamate and so forth. Moreover, compounds with FD < 27 were considered to make only a minor contribution to the overall aroma. Compared with A, sample B presented anethole (anise, slight sweet) and unknown 1 (almond, caramel, toasty) with the highest aroma intensity (FD = 243). For sample B, Limonene (fruity, orange) and (*Z*)-cinnamaldehyde (cinnamon) gave the higher FD factor of 81. However, 3-phenylpropanal, 2-methoxybenzaldehyde, β-pinene, 1-terpinen-4-ol, acetoin, and unknown1 had no FD factor in sample A; and nonanal, (*Z*)-cinnamaldehyde, γ-undecalactone, coumarin, and anethole showed a lower FD factor in sample A. Therefore, the two samples exhibited different aromas in sensory evaluation. 

However, owing to the strong complexity of the analyte, which led to very rich chromatographic profiles and made it quite difficult to sniff, the evaluation of a single peak or a given chromatographic region was often affected by the former eluents. Thus, odor descriptions of the sniffers were prone to differ from the literatures. Anyway, some compounds have been previously identified as key odor-active compounds in fried bacon, fried pork loin, and roasted pork of mini-pig by GC–O [29,30], for instance pentanal, limonene, furfural, β-pinene, and anethole. For the present work, spice ingredients (onion, ginger, garlic, cinnamon, and star aniseed) were used in the cooking, and thus, a high amount of spice compounds, e.g. (Z)-cinnamaldehyde, β-pinene, 1-terpinen-4-ol, anethole, α-pinene, δ-3-carene, 1,8-cineole, ethyl cinnamate were contained in the volatiles of the braised pork; they were basically similar to the results in Reference [31].

### 3.4. Quantitation of Important Odorants and Calculation of OAV

It is well known that AEDA is a worthy method for the screening of odor-active compounds in a given food. However, the application of AEDA does not provide immediate information about the contribution of a single odorant to the overall aroma. To get a deeper insight into the contribution of the quantitated aroma compounds to the overall aroma of the braised pork, OAV was calculated for each aroma component. Moreover, OAV was usually used to provide a rough evaluation of the real contribution of each compound to the overall aroma.

For quantification purposes, calibration curves for each odor-active compound in the extracts of two types of braised pork by SDE-AEDA-GC/O were drawn (Table 4). From Table 4, the coefficient of determination was determined, which indicated strong linearity for each of the standards.

Table 3 lists the concentrations (calculated by the standard curves), odor threshold values, and calculated OAV of the odor potent compounds. In terms of OAV, 3-methylbutanal, pentanal, nonanal, (E,E)-2,4-decadienal, phenylacetaldehyde, dodecanal, and linalool had OAV values >200, indicating that these compounds, especially aldehydes, might significantly contribute to the overall aroma of both types of braised pork. These results are consistent with the fact that aldehydes have an important potential effect on the global flavor of meat species [32]. By comparison, sample A had a higher content of (E,E)-2,4-decadienal (OAV = 1304). Furthermore, many compounds such as (*Z*)-cinnamaldehyde, limonene, γ-undecalactone, anethole, and 1,8-cineole were deemed to be the key-aroma compounds in braised pork, due to their concentrations clearly exceeding their odor thresholds. It is worth mentioning that two unknown compounds cannot calculate the OAV, so they can only be preliminarily considered as potential aroma-active substances through the odor description and FD factor. In addition, for some compounds, the concentrations did not reach their odor thresholds and they showed an OAV of <1. However, the combination of these compounds might also be linked with the aroma of braised pork due to the synergistic effect of similar compounds. Lack of agreement between FD factor and OAV results also existed in braised pork; 3-Methylbutanal had a high OAV (OAV = 335, 211) but a low FD factor (FD = 3), while α-Pinene had a high FD factor but a low OAV, which showed the influence of the food matrix.

## 4. Conclusions

Considering that changes in eating habits are difficult to achieve, strategies that do not require consumers’ willpower to change have the greatest chance of succeeding at the population level in the short term. Thus, reformulation of products has been proposed as one of the most effective strategies to encourage changes in nutrient intake. The main challenge for reducing the added sugar content of food products is that it causes changes in their sensory characteristics (especially aroma and taste attributes), which are key determinants of consumers’ liking. In this study, the novel braised sauce without white granulated sugar added was tested to see if it had the same aroma perception as white granulated sugar. Descriptive sensory analysis showed that the effect of self-made braised sauce in braised pork was as good as white granulated sugar. One-hundred-and-nine volatile compounds were identified by GC-MS using SPME and SDE methods in two braised porks. From the SDE-AEDA-GC/O analysis, it was found that pentanal (almond, pungent), nonanal (fat, green), ((E, E)-2,4-decadienal (fat, roast), phenyl acetaldehyde (hawthorne, honey, sweet), dodecanal (lily, fat, citrus) and linalool showed the highest OAV values (>200), indicating a contribution to the aroma of braised pork. The novel self-made braised sauce was proved to be useful in cooking braised pork with good sensory characteristics and rich aroma compounds. On the other hand, it is easy and convenient to operate for ordinary consumers; therefore, the braised sauce has significance in research and development.

Although people do not eat braised pork every day, at least the same aroma effect can be achieved by sugar substitution, which is a pleasure for people who want to eat but have to control their sugar intake. In addition, further research in this respect should be carried out on how many calories are saved when sugar is reduced and the effect of the novel formula on people’s health.

## Figures and Tables

**Figure 1 foods-08-00087-f001:**
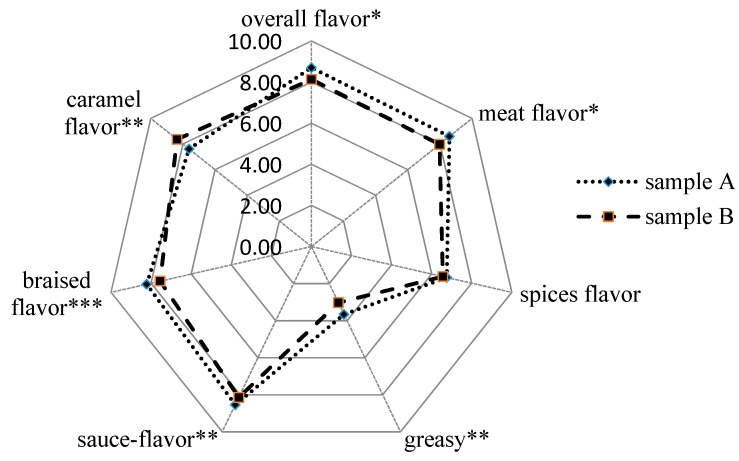
Sensory evaluation results of two braised pork samples (* means significant at *p* < 0.05 level, ** means significant at *p* < 0.01 level, *** means significant at *p* < 0.001 level; sample A: prepared from self-made braised sauce; sample B: prepared from white granulated sugar).

**Figure 2 foods-08-00087-f002:**
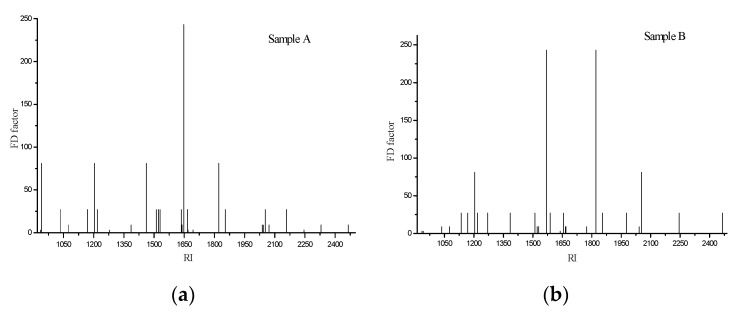
(**a**) FD chromatogram of the volatile fraction of braised pork with braised sauce; (**b**) FD chromatogram of the volatile fraction of braised pork with white granulated sugar. (FD: Flavor Dilution factor, RI: retention indices).

**Table 1 foods-08-00087-t001:** Identification of braised pork volatile compounds extracted using SPME-GC/MS.

Compounds	ID ^X^	RI ^Y^	RI ^Z^	Peak Area (%)	
				A	B
Hydrocarbons					
α-Pinene	M	1036	1032 ^m^	0.29 ± 0.07	0.40 ± 0.09
Camphene	M	1078	1075 ^m^	0.90 ± 0.02	1.04 ± 0.05
β-Pinene	M	1135	1116 ^m^	0.04 ± 0.01	0.06 ± 0.02
δ-3-Carene	M	1169	1148 ^m^	0.04 ± 0.01	nd
β-Myrcene	M	1171	1145 ^m^	0.12 ± 0.03	0.09 ± 0.04
α-Phellandrene	M	1176	1166 ^m^	nd	0.05 ± 0.02
Limonene	M	1204	1201 ^m^	0.32 ± 0.11	0.22 ± 0.08
Bornylene	S	1217		nd	0.05 ± 0.01
β-Phellandrene	M	1214	1209 ^m^	0.37 ± 0.12	0.38 ± 0.06
Styrene	M	1272	1241 ^m^	0.07 ± 0.01	0.23 ± 0.11
o-Cymene	S	1280		0.04 ± 0.01	0.06 ± 0.02
Total				2.19	2.59
Aldehydes					
2-Methylpropanal	M	834	821 ^m^	0.09 ± 0.01	0.07 ± 0.01
3-Methylbutanal	M	935	910 ^m^	nd	0.80 ± 0.21
2-Methylbutanal	M	933	912 ^m^	0.48 ± 0.13	nd
Pentanal	M	942	935 ^m^	0.13 ± 0.01	0.17 ± 0.02
Hexanal	M	1076	1084 ^m^	0.25 ± 0.02	0.63 ± 0.14
Heptanal	M	1174	1174 ^m^	0.08 ± 0.03	0.08 ± 0.02
Octanal	M	1303	1280 ^m^	0.04 ± 0.01	0.04 ± 0.01
(E)- 2-Heptenal	M	1344	1324 ^g^	0.11 ± 0.04	0.05 ± 0.01
Nonanal	M	1385	1385 ^m^	0.12 ± 0.02	0.15 ± 0.03
Benzaldehyde	M	1522	1495 ^m^	0.49 ± 0.16	0.57 ± 0.13
Phenylacetaldehyde	M	1639	1625 ^m^	0.06 ±0.02	0.08 ± 0.01
(E,E)-2,4-Decadienal	M	1654	1632 ^m^	0.04 ± 0.01	0.02 ± 0.01
Geranial	M	1745	1715 ^m^	0.03 ± 0.01	0.03 ± 0.01
4-Methoxybenzaldehyde	S	2042		0.04 ± 0.01	0.03 ± 0.01
(*Z*)-Cinnamaldehyde	S	2053		0.68 ± 0.13	0.28 ± 0.04
Total				2.64	3.00
Alcohols					
Ethanol	M	959	929 ^m^	0.52 ± 0.13	0.80 ± 0.21
Cyclopentanol	S	1004		nd	0.14 ± 0.03
Allyl alcohol	S	1141		nd	0.06 ± 0.01
1-Butanol	M	1169	1145 ^m^	0.06 ± 0.01	nd
3-Methyl-1-butanol	M	1228	1205 ^m^	nd	0.07 ± 0.01
1-Pentanol	M	1234	1255 ^m^	0.08 ± 0.02	0.10 ± 0.04
1-Hexanol	M	1373	1360 ^m^	nd	0.03 ± 0.01
1-Terpinen-4-ol	M	1589	1591 ^m^	0.02 ± 0.01	0.03 ± 0.01
2-(2-Ethoxyethoxy)ethanol	S	1650		0.02 ± 0.01	0.08 ± 0.02
α-Terpineol	M	1681	1688 ^m^	0.02 ± 0.01	0.03 ± 0.01
4-Hydroxy-4-methyl-2-pentanone	S	2350		nd	0.07 ± 0.03
Total				0.72	1.40
Ketones					
2-Pentanone	M	1002	983 ^m^	0.12 ± 0.03	nd
Acetoin	M	1270	1287 ^m^	0.10 ± 0.02	0.07 ± 0.02
6-Methyl-5-hepten-2-one	M	1357	1336 ^m^	0.06 ± 0.02	nd
Total				0.28	0.07
Acids					
Acetic acid	M	1445	1450 ^m^	0.30 ± 0.12	0.20 ± 0.09
Propanoic acid	M	1551	1523 ^m^	0.04 ± 0.02	nd
Butanoic acid	M	1610	1619 ^m^	0.04 ± 0.01	0.06 ± 0.02
Hexanoic acid	M	1892	1888 ^g^	0.03 ± 0.01	0.04 ± 0.01
Decanoic acid	M	2333	2361 ^m^	0.15 ± 0.07	nd
Total				0.57	0.30
Esters and lactones					
Methyl butanoate	S	1011		nd	0.07 ± 0.02
γ-Butyrolactone	M	1630	1647 ^m^	0.03 ± 0.01	nd
Styrallyl acetate	S	1737		0.03 ± 0.01	0.05 ± 0.01
Benzyl isovalerate	S	1938		nd	0.03 ± 0.01
Total				0.06	0.14
Phenols					
Phenol	S	2072		0.02 ± 0.01	0.03 ± 0.01
Total				0.02	0.03
Ethers					
Estragole	M	1656	1655 ^m^	nd	0.04 ± 0.01
Anethole	S	1821		1.31 ± 0.05	1.32 ± 0.03
Total				1.31	1.36
Sulphur compounds					
Methanethiol	M	711	696 ^m^	2.87 ± 0.11	3.12 ± 1.01
Allyl mercaptan	S	916		0.34 ± 0.06	1.31 ± 0.08
Allyl methyl sulfide	S	981		0.07 ± 0.01	0.10 ± 0.02
2-Methylthiophene	M	1110	1084 ^m^	nd	0.06 ± 0.01
Diallyl sulfide	S	1165		nd	0.11 ± 0.07
Thiophene	M	1178	1150 ^m^	0.02 ± 0.01	nd
Thiazole	M	1271	1286 ^g^	0.02 ± 0.01	nd
Allyl methyl disulfide	S	1298		nd	0.07 ± 0.02
2-Methyl-3-furanthiol	M	1310	1339 ^g^	0.03 ± 0.01	nd
Furfuryl mercaptan	M	1431	1432 ^m^	0.05 ± 0.01	nd
Methional	M	1484	1458 ^m^	0.16 ± 0.02	nd
2-Acetylthiazole	M	1686	1692 ^g^	0.08 ± 0.03	0.10 ± 0.03
Total				3.65	4.87
Nitrogenous compounds					
Pyrazine	M	1235	1254 ^g^	0.05 ± 0.01	nd
Methanamide	S	1605		0.02 ± 0.01	nd
2-Amino-6-methylbenzoic acid	S	1781		nd	0.20 ± 0.09
2-Acetylpyrrole	M	2037	2027 ^g^	0.02 ± 0.01	nd
Dimethylamine	S	2232		0.01 ± 0.00	0.11 ± 0.04
Total				0.11	0.31
Oxygen-containing heterocycles					
2-Ethylfuran	M	979	986 ^g^	nd	0.05 ± 0.01
2-Pentylfuran	M	1243	1240 ^m^	0.10 ± 0.03	0.12 ± 0.05
2-Methyltetrahydrofuran-3-one	M	1278	1299 ^g^	0.12 ± 0.04	nd
Furfural	M	1461	1455 ^m^	0.22 ± 0.11	0.04 ± 0.01
5-Methyl-2-furfural	M	1570	1560 ^m^	0.06 ± 0.01	nd
Furfuryl alcohol	S	1647		0.03 ± 0.00	nd
2,3-Dihydro-3,5-dihydroxy-6-methyl-4H-pyran-4-one	S	2359		0.24 ± 0.10	nd
5-Hydroxymethyl-2-furfural	S	2512		0.13 ± 0.03	nd
Total				0.91	0.20

The data is mean ± standard deviation of triplicate analysis. nd: not found. X: Identification method (ID): S, mass spectrum and RI agree with that of the authentic compound run under similar GC-MS conditions; M, mass spectrum and RI agree with literature data. Y: Linear retention indices (RI) calculated of unknown compounds on a HP-Wax (HP-PEG-INNOWAX) (60 m × 0.25 mm × 0.25µm) with a homologous series of n-alkanes (C6-C30). Z: m: RI from the flavornet database [16]; g: references [17,18]; A: sample A prepared from self-made braised sauce; B: sample B prepared from white granulated sugar. SPME: solid phase microextraction; GC: gas chromatography; MS: mass spectrometry; ID: Identification method; RI: retention indices.

**Table 2 foods-08-00087-t002:** Identification of braised pork volatile compounds extracted using SDE-GC/MS.

Compounds	ID ^X^	RI ^Y^	RI ^Z^	Peak Area (%)	
				A	B
Hydrocarbons					
α-Pinene	M	1036	1032 ^m^	0.03 ± 0.01	0.03 ± 0.01
β-Pinene	M	1135	1116 ^m^	nd	0.17 ± 0.04
δ-3-Carene	M	1169	1148 ^m^	0.03 ± 0.01	0.17 ± 0.05
β-Myrcene	M	1171	1145 ^m^	0.07 ± 0.01	nd
Limonene	M	1204	1201 ^m^	0.15 ± 0.04	0.30 ± 0.12
o-Cymene	S	1280		0.01 ± 0.00	nd
Total				0.29	0.66
Aldehydes					
3-Methylbutanal	M	935	910 ^m^	0.01 ± 0.01	0.01 ± 0.01
Pentanal	M	942	935 ^m^	0.64 ± 0.24	nd
Hexanal	M	1076	1084 ^m^	0.31 ± 0.11	0.05 ± 0.02
Heptanal	M	1174	1174 ^m^	0.23 ± 0.04	nd
Nonanal	M	1385	1385 ^m^	0.12 ± 0.03	0.14 ± 0.01
Benzaldehyde	M	1522	1495 ^m^	0.06 ± 0.03	1.18 ± 0.05
(E)-2-Decenal	S	1635		0.01 ± 0.00	0.01 ± 0.01
Phenyl acetaldehyde	M	1639	1625 ^m^	0.19 ± 0.01	0.37 ± 0.06
2,4-Decadienal	M	1654	1632 ^m^	nd	0.01 ± 0.01
(E)-Cinnamaldehyde	M	1666	1631 ^m^	0.23 ± 0.08	0.11 ± 0.04
Neral	M	1669	1667 ^m^	0.10 ± 0.08	0.04 ± 0.01
Dodecanal	M	1693	1722 ^m^	0.13 ± 0.06	nd
Benzenepropanal	S	1775		0.02 ± 0.01	0.46 ± 0.12
(E,E)-2,4-Decadienal	S	1854		0.02 ± 0.01	nd
2-Methoxybenzaldehyde	S	1977		nd	0.28 ± 0.05
4-Methoxybenzaldehyde	S	2042		0.05 ± 0.01	0.34 ± 0.11
(*Z*)-Cinnamaldehyde	S	2053		0.74 ± 0.16	1.09 ± 0.41
(Z)-13-Octadecenal	S	2383		nd	0.39 ± 0.11
2-Methoxycinnamaldehyde	S	2401		nd	0.34 ± 0.09
Total				2.85	4.80
Alcohols					
Allyl alcohol	S	1105		0.35 ± 0.02	nd
1,8-Cineole	M	1218	1213 ^m^	0.08 ± 0.01	0.13 ± 0.02
1-Pentanol	M	1234	1255 ^m^	0.08 ± 0.01	nd
Linalool	M	1530	1537 ^m^	0.07 ± 0.02	0.44 ± 0.11
1-Terpinen-4-ol	M	1589	1591 ^m^	nd	0.08 ± 0.02
α-Terpineol	M	1681	1688 ^m^	0.22 ± 0.10	0.08 ± 0.02
β-Fenchyl alcohol	S	1701		nd	0.05 ± 0.01
Cinnamyl alcohol	M	2330	2300 ^m^	0.15 ± 0.06	nd
(E)-2-Tetradecen-1-ol	S	2356		nd	0.26 ± 0.08
Total				0.96	1.04
Ketones					
2,3-Pentanedione	S	1057		0.10 ± 0.04	nd
2-Heptanone	M	1172	1170 ^m^	0.01 ± 0.00	nd
2-Octanone	M	1267	1244 ^m^	0.04 ± 0.01	nd
Acetoin	M	1270	1287 ^m^	nd	0.21 ± 0.07
Acetol	M	1286	1287 ^m^	nd	0.09 ± 0.02
2-Tridecanone	S	1804		0.01 ± 0.01	nd
2-Pentadecanone	S	2020		0.06 ± 0.01	0.35 ± 0.11
1-(4-Methoxyphenyl)-2-propanone	S	2170		0.01 ± 0.00	nd
2-Heptadecanone	S	2208		0.01 ± 0.01	nd
Total				0.24	0.65
Acids					
Butanoic acid	M	1610	1619 ^m^	0.01 ± 0.01	nd
Heptanoic acid	S	1935		0.01 ± 0.01	nd
Tetradecanoic acid	M	2094	2094 ^g^	0.05 ± 0.02	nd
Nonanoic acid	M	2193	2202 ^m^	0.01 ± 0.01	nd
Decanoic acid	M	2333	2361 ^m^	0.03 ± 0.01	nd
Oleic acid	M	2374	2430 ^m^	0.05 ± 0.02	nd
Hexadecanoic acid	S	2468		0.16 ± 0.07	0.46 ± 0.11
Total				0.32	0.46
Lactones and esters					
γ-Butyrolactone	M	1630	1647 ^m^	0.11 ± 0.03	nd
γ-Nonalactone	M	2014	2042 ^m^	0.06 ± 0.01	nd
γ-Undecalactone	M	2145	2270 ^m^	0.27 ± 0.06	1.58 ± 0.31
Ethyl cinnamate	M	2158	2139 ^m^	0.01 ± 0.01	nd
Total				0.45	1.58
Phenols					
Phenol	S	2072		0.78 ± 0.11	1.05 ± 0.12
Total				0.78	1.05
Ethers					
Estragole	M	1656	1655 ^m^	0.02 ± 0.01	0.98 ± 0.09
Anethole	S	1821		1.78 ± 0.08	1.16 ± 0.21
Total				1.80	2.15
Nitrogenous compounds					
2-Acetylpyrrole	M	2037	2025 ^g^	0.02 ± 0.01	nd
Total				0.02	
Oxygen-containing heterocycles					
Furfuryl alcohol	M	1157	1199 ^m^	0.07 ± 0.02	nd
Furfural	M	1461	1455 ^m^	0.06 ± 0.02	nd
2-Acetylfuran	M	1511	1490 ^m^	0.01 ± 0.01	0.01 ± 0.01
5-Methyl-2-furfural	M	1570	1560 ^m^	nd	0.05 ± 0.02
5-Methyl-2-furfuryl alcohol	S	1700		0.03 ± 0.01	nd
3,4-Dihydro- 2H-1-benzopyran	S	2275		nd	0.33 ± 0.13
2-Methyltetrahydrofuran-3-one	S	2393		0.02 ± 0.01	nd
Coumarin	M	2465	2465 ^m^	0.03 ± 0.01	0.31 ± 0.12
Total				0.22	0.70

The data is mean ± standard deviation of triplicate analysis. nd: not found. X: Identification method (ID): S, mass spectrum and RI agree with that of the authentic compound run under similar GC-MS conditions; M, mass spectrum and RI agree with literature data. Y: Linear retention indices (RI) calculated of unknown compounds on a HP-Wax (60 m × 0.25 mm × 0.25 µm) with a homologous series of n-alkanes (C6-C30). Z: m: RI from the flavornet database [16]; g: references [17,18]. A: sample A prepared from self-made braised sauce; B: sample B prepared from white granulated sugar; SDE: simultaneous distillation and extraction.

**Table 3 foods-08-00087-t003:** Odor-active compounds detected in in the SDE extracts of two types of braised pork by AEDA.

RI ^a^	Compounds	Odor Description ^b^	Odor Threshold (μg/g)	A			B		
				CN ^c^	FD ^d^	OAV ^e^	CN	FD	OAV
Aldehydes									
935	3-Methylbutanal	burnt-sweet, roast	0.001	0.335	3	335	0.211	3	211
942	Pentanal	almond, pungent	0.009	3.0012	81	333	0.1045	3	12
1076	Hexanal	grass, tallow, fat	0.0045	1.4725	9	34	0.2148	9	5
1385	Nonanal	fat, green	0.0011	0.5744	9	522	0.5637	27	512
1461	Furfural	bread, almond, sweet	3	0.3468	81	<1			
1522	Benzaldehyde	almond, burnt sugar	0.35	0.3224	27	<1	4.6550	9	13
1635	(E)-2-Decenal	grass, earthy	0.0004	0.0735	27	180			
1639	Phenylacetaldehyde	hawthorne, honey, sweet	0.004	0.8829	9	220	1.4606	3	365
1666	(E)-Cinnamaldehyde	cinnamon, paint	−f	1.0826	27		0.4173	9	
1669	Neral	lemon	0.053	0.4836	3	9	0.1457	9	3
1693	Dodecanal	lily, fat, citrus	0.002	0.6133	3	307			
1775	3-Phenylpropanal	grass, fat	0.12				1.8005	9	9
1854	(E,E)-2,4-Decadienal	fat, roast	0.00007	0.0913	27	1304	0.0237	27	339
1977	2-Methoxybenzaldehyde	wax, medicinal	-				1.1077	27	
2042	4-Methoxybenzaldehyde	similar hawkthorn	0.03	0.2233	9	7	1.3564	9	45
2053	(*Z*)-Cinnamaldehyde	cinnamon	0.16	3.4719	27	22	4.2745	81	27
Hydrocarbons									
1036	α-Pinene	green, fresh	0.006	0.1321	27	22	0.2148	9	36
1135	β-Pinene	mild, green	0.14				0.6577	27	5
1169	δ-3-Carene	lemon, resin	0.77	0.1547	27	<1	0.6634	27	<1
1204	Limonene	fruity, orange	0.01	0.7133	81	71	1.1866	81	119
Alcohols									
1530	Linalool	floral, lavender	0.006	0.3115	27	52	1.7199	9	287
1589	1-Terpinen-4-ol	turpentine, nutmeg, must	1.29				0.3126	27	<1
2330	Cinnamyl alcohol	oily	0.077	0.7314	9	9			
Ketones									
1270	Acetoin	butter, cream	0.8				0.8223	27	1
Esters and lactones									
2245	γ-Undecalactone	flower, wax	0.042	1.2843	3	31	6.2292	27	148
2158	Ethyl cinnamate	cinnamon, honey	0.04	0.0734	27	2			
Phenols									
2072	Phenol	phenol	5.9	3.6822	9	<1			
Ethers									
1218	1,8-Cineole	mint, sweet	0.012	0.4129	27	34	0.5122	27	43
1821	Anethole	anise, slight sweet	0.16	8.3842	81	52	4.5618	243	29
Nitrogenous compounds									
2037	2-Acetylpyrrole	nut, walnut, burnt	170	0.0934	9	<1			
Oxygen-containing heterocycles									
1278	2-Methyltetrahydrofuran-3-one	sweet, caramel	-	0.0937	3				
1511	2-Acetylfuran	butter, meaty	0.01	0.0429	27	4	0.0419	27	4
1647	Furfuryl alcohol	sauce-flavor	2	0.3126	243	<1			
2465	Coumarin	green, sweet	0.33	0.1622	9	<1	1.2419	27	4
Unknown									
1570	Unknown1	almond, caramel, toasty	-					243	
1656	Unknown2	licorice, anise	-		27			27	

a: Linear retention indices calculated of unknown compounds on a HP-Wax (60 m × 0.25 mm × 0.25 µm) with a homologous series of n-alkanes (C6-C30). b: Odor description: odor description detected and described by panelists during olfactometry. c: Concentration: Results are as μg/g, calculated by the standard curves in Table 4. d: FD factor is the highest dilution of the extract at which an odorant is determined by aroma extract dilution analysis. e: OAV (odor activity values), concentration divided by odor threshold. f: Odor thresholds were unavailable. CN: concentration; FD: Flavor Dilution factor. AEDA: aroma extract dilution analysis.

**Table 4 foods-08-00087-t004:** Standard curves of key aroma compounds in braised pork.

No	Compounds	Quantitative Ions ^a^	R^2^	LOD (μg/kg) ^b^	LOQ (μg/kg) ^c^
1	3-Methyl- butanal	41,43,44	0.987	1.8	6.0
2	Pentanal	41,44,58	0.974	4.4	14.7
3	α-Pinene	91,92,93	0.992	2.0	6.7
4	Hexanal	44,56,57	0.978	25.1	83.6
5	β-Pinene	69,91,93	0.988	1.5	5.0
6	δ-3-Carene	77,91,93	0.995	6.4	21.3
7	Limonene	67,68,93	0.996	1.9	6.3
8	1,8-Cineole	43,81,108	0.995	1.6	5.3
9	Acetoin	43,45,88	0.996	6.2	20.6
10	2-Methyltetrahydrofuran-3-one	43,72,100	0.978	2.1	7.0
11	Nonanal	41,56,57	0.986	7.0	23.3
12	Furfural	39,95,96	0.986	10.6	35.3
13	2-Acetylfuran	39,95,110	0.976	1.2	4.0
14	Benzaldehyde	77,105,106	0.985	16.3	54.3
15	Linalool	55,71,93	0.971	4.3	14.3
16	1-Terpinen-4-ol	71,93,111	0.983	1.5	5.0
17	(E)-2-Decenal	41,55,70	0.977	0.3	1.0
18	Phenyl acetaldehyde	69,91,92	0.972	4.3	14.3
19	Furfuryl alcohol	81,97,98	0.994	6.8	22.6
20	(E)-Cinnamaldehyde	103,131,132	0.964	3.2	10.7
21	Neral	41,69,94	0.979	11.8	39.3
22	Dodecanal	41,57,82	0.986	2.2	7.3
23	3-Phenylpropanal	91,92,134	0.974	2.4	8.0
24	Anethole	132,133,148	0.995	1.7	5.7
25	(E,E)-2,4-Decadienal	41,67,81	0.995	0.8	2.7
26	2-Methoxybenzaldehyde	77,118,136	0.986	5.3	17.6
27	2-Acetylpyrrole	66,94,109	0.976	1.7	5.7
28	4-Methoxybenzaldehyde	77,92,135	0.989	14.6	48.6
29	(*Z*)-Cinnamaldehyde	103,131,132	0.977	0.6	2.0
30	Phenol	65,66,94	0.994	5.4	18.0
31	Ethyl cinnamate	103,131,176	0.974	1.6	5.3
32	γ-Undecalactone	55,85,128	0.983	2.1	7.0
33	Cinnamyl alcohol	91,92,134	0.986	0.6	2.0
34	Coumarin	89,118,146	0.976	2.1	7.0

^a^ Selected ions used for quantitation. ^b^ LOD, limits of detection. ^c^ LOQ, limits of quantitation.

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
