# Peer review of "Aroma Patterns Characterization of Braised Pork Obtained from a Novel Ingredient by Sensory-Guided Analysis and Gas-Chromatography-Olfactometry"

_foods, 2019, doi:10.3390/foods8030087_

Round 1
Reviewer 1 Report
The authors studied the difference on volatile compounds between the braised pork with self-made braised sauce and the traditional braised pork by GC-MS. In addition, the odor-active compounds in these samples were identified by GC-O, AEDA, and the OAV calculation. Furthermore, the sensory evaluation was carried out.
The authors carefully carried out the experiments in sensory evaluation and AEDA. However, the SDE experiment was incompatible. Further, the significant revision is necessary.
Abstract
Two types of braised pork were prepared from self-made braised sauce added to Maillard reaction intermediate (MRI) and white granulated sugar, respectively. Descriptive sensory analysis and gas chromatography-mass spectrometry (GC-MS) were conducted to investigate their differences in sensory and aroma compounds. The results showed that the effect of self-made braised sauce in braised pork was comparable to the white granulated sugar. One hundred nine volatile flavor compounds were identified by GC-MS using headspace-solid phase microextraction (HS-SPME) and simultaneous distillation and extraction (SDE). Thirty-six odor active compounds with retention index ranged from 935–2465 were identified by aroma extract dilution analysis (AEDA). Additionally, their odor activity values (OAV) were calculated. It was found that 17 aroma compounds showed an OAV greater than 1. Among them, pentanal (almond, pungent), nonanal (fat, green), ((E, E)-2,4-decadienal (fat, roast), phenylacetaldehyde (hawthorn, honey, sweet), dodecanal (lily, fat, citrus) and linalool (floral, lavender) reached the highest OAV values (>200), indicating a significant contribution to the aroma of two types of braised pork. These results indicated that the self-made braised sauce added with MRI could be used for cooking braised pork with good sensory characteristics.
Double underline: Because there is not result obtained by multivariate analysis (PLS etc.), this sentence is unconfirmed.
Underline: These are my correction.
1.Introduction
Line 50: xylose-l-cysteine
Line 51: gas chromatography-olfactometry
2.Materials and Methods
Line 68: 3-Methylbutanal
Line 70: phenylacetaldehyde
Lines 70 and 72: Were “(E)-cinnamaldehyde” and “cinnamaldehyde” different compounds?
Line 71: 3-phenylpropanal
Line 72: 4-Methoxybenzaldehyde
Line 73: Please delete “. All of the standard chemicals above”. “…and coumarin were purchased from …”
Line 74: Please delete “and were of GC quality”.
Line 78: Please replace “in n-alkanes” by “in hexane”.
Line 81: Please describe the location of Anhui Muyang Oil and Fats Co., Ltd.
Line 82: 2.3. Preparation of self-made braised sauce
Lines 83-84: Xylose (4.5 g) and l-cysteine (0.54 g)(molar ratio about 10:1.5) were dissolved in water (50 g).
Lines 82-94 and 94-106: This preparation should be carried out at least in triplicate because the recipe is extremely complex.
Line 113: In Results, it is better to show the scores obtained (means and standard deviation) in a table.
Line 125: “caramel” flavor
Lines 125-126: In Results, the scores obtained (means and standard deviation) should be shown in a table.
Lines 133-134: Please delete “at 250°C for 5 min”.
Line 135-145: The authors should carry out the SAFE procedure instead of the SDE procedure.
Line 139: under nitrogen atmosphere?
Line 140: collect in dichloromethane
Line 141: Na2SO4
Line 144: The weight (g) of the extracts obtained should be shown for the quantitation
Line 163: The split ratio was 10:1. The eluate was split…
Line 183: Zero point zero one
Line 185: Were the ratios 1:10 (1/11)-1:60 (1/61) equal to 10-fold-60-fold dilutions?
Lines 191-194: Were the concentrations the values in extract or braised pork?
Line 198: detection threshold concentration in water (μg/g)
3. Results
3.1. Sensory Evaluation of the Braised Pork Samples
As mentioned above, the authors should show the sores (means and standard deviation) in a table. If possible, please also show the spider charts in a figure. There is no need for Figure 1 and the table on F-values.
Lines 231-233: The reason is not correct because the volatile compounds, such as furfural, 5-methylfurfural, 2,3-dihydro-3,5-dihydroxy-6-methyl-4H-pyran-4-one etc., were generated in sample A in larger amounts than in sample B. These compounds might be generated via caramelization of sugars.
3.2 Volatile Flavor Compounds Identified from the Braised Pork
Lines 243-244: The authors mentioned the advantages of the SPME method. Among these, there is the advantage such as “the volatile compounds detected do not change”. The authors should have been paying attention to the change of volatile compounds. Therefore, the authors should not apply the SDE method.
Lines 248-249: “All 109 volatile…significantly different (p <0.05)”. The authors should show the evidences. The authors should confirm the correlations between sensory attributes and volatile flavor compounds by multivariate analysis.
Line 249: Seventy-seven
Line 252: “The difference…principle and conditions”. This reason is too common.
Lines 257-259: The authors should not apply the SDE method. The sulfur-containing compounds such as furfuryl mercaptan and 2-methyl-3-furanthiol may be contributors to the overall aroma of meat samples.
Line 260: “that might lead to…some volatile compounds”. The authors should apply the SAFE method instead of the SDE method.
Table 2
o-Cymene: p-Cymene?
2-Methyl-propanal: 2-Methylpropanal
3-Methyl-butanal: 3-Methylbutanal
2-Methyl-butanal: 2-Methylbutanal
Phenyl acetaldehyde: Phenylacetaldehyde
2,4-Decadienal: (E,E)-2,4-Decadienal
Anisic aldehyde: 4-Methoxybenzaldehyde
Ethyl digol: 2-(2-Ethoxyethoxy)ethanol
Diacetone alcohol: 4-Hydroxy-4-methyl-2-pentanone
Methylisohexenyl ketone: 6-Methyl-5-hepten-2-one
Methyl butyrate: Methyl butanoate
γ-Butyrolactone: category of “Lactone”
Estragol: Estragole
2-Propene-1-thiol: Allyl mercaptan
Methyl allyl sulfide: Allyl methyl sulfide
Methylthiophene: 2 or 3-Methylthiophene?
Allyl sulfide: Diallyl sulfide
2-Methyl-3-sulfanylfuran: 2-Methyl-3-furanthiol
2-Acetyl thiazole: 2-Acetylthiazole
5-Methylfurfural: 5-Methyl-2-furfural
5-Hydroxymethylfurfural: 5-Hydroxymethyl-2-furfural
Table 3
o-Cymene: p-Cymene?
3-Methyl- butanal: 3-Methylbutanal
(E)-Cinnamaldehyde and Cinnamaldehyde: different compounds?
Lauric aldehyde: Dodecanal
Anisic aldehyde: 4-Methoxybenzaldehyde
E-2-Tetradecen-1-ol: (E)-2-Tetradecen-1-ol
Heptanone: 2, 3, or 4-Heptanone ?
Hydroxyacetone: Acetol
γ-Nonanolactone: γ-Nonalactone
Ethyl cinnamate: category of “Ester”
2,6-Di-tert-butyl-4-methylphenol and 2,4-Di-tert-butylphenol: These compounds are food additives as antioxidant. Please delete these compounds.
Estragol: Estragole
5-Methylfurfural: 5-Methyl-2-furfural
line281: Are “the semi-quantitation results” GC peak area percentages?
Lines 283-287: These sentences should be deleted because of the discussion according to the later AEDA and OAV experiments.
Line 288: 3-methyl-butanal: 3-methylbutanal
Line 289: phenyl acetaldehyde: phenylacetaldehyde
Line 289: 2,4-decadienal: (E,E)-2,4-decadienal
Line 289: anisic aldehyde: 4-methoxybenzaldehyde
Line 290: cinnamaldehyde: (E)-cinnamaldehyde or cinnamaldehyde?
Lines 296-298: This reason is not correct because the major sulfur compounds are allyl mercaptan derivatives which are not generated via Maillard reaction in meat but via degradation of the precursors in garlic material.
Line 301: This reason is incompatible with the sentence described at lines 243-244. Further, the SPME procedure is carried out in a sealing vial. Therefore, the loss of volatiles might be easily occurred.
Line 302: Was leucine derived from the pork meat?
Line 303: Please delete “and methionine”.
Line 303: “It can provide… fruity odor characteristics”. This is not correct. Please delete.
Lines 304-305: 3-Methylbutanal may not contribute to the caramel odor. This sentence is not correct. Please delete.
Lines 308-309: Please delete the sentences because the thiophene compounds were not important odorants.
Line 316: phenyl acetaldehyde: phenylacetaldehyde
Line 316: benzenepropanal: 3-phenylpropanal
Line 317: only one lactone γ-undecalactone was detected
Lines 317-319: This is not correct. Please delete.
3.3. Identification of the Odor-Active Compounds in Braised Pork
Line 321: in braised pork: in the extracts of two types of braised pork
Lines 321-322: Please delete “by SDE-AEDA-GC-MS/O”. Please replace by “by AEDA”.
Line 323: 1 ketone
Line 323: 1 phenol
Line 323-324: 1 nitrogenous compound
Line 324: Thirty
Line 327: RI 1135-2053
Line 327: Furfuryl alcohol does not have a “meaty” odor.
Line 329-331: The authors mentioned the potent odorants in “braised pork”. However, the AEDA results are limited to the SDE extracts. Therefore, this sentence is not correct.
Line 332: Table 4 Odor-active compounds detected in the SDE extracts of two types of braised pork by AEDA
Table 4
I think that maltol, furaneol, homofuraneol, sotolone, and abhexone, which are Maillard reaction (or caramelization) products, may be the contributors because of having a low odor threshold. However, these compounds were not determined by GC-O and AEDA in this study. The reason is the application of the SDE method.
(E)-Cinnamaldehyde and Cinnamaldehyde: different compounds?
Phenyl acetaldehyde: Phenylacetaldehyde
Lauric aldehyde: Dodecanal
Anisic aldehyde: 4-Methoxybenzaldehyde
γ-Undecalactone: category of “Lactone”
In the column of Odor Threshold and Line 340: Were all values measured? Why can you measure the values of (E)-cinnamaldehyde, 2-methoxybenzaldehyde, and 2-methyltetrahydrofuran-3-one? These standard compounds can be obtained from the suppliers.
Line 336: Do the concentrations indicate the value in extracts or in pork?
Line 341 and 342: Please replace “cooked by” by “with”.
Lines 343-346: Please delete.
Lines 352-353: The addition of spices is a common operation for sample A and B. The sentence is not reason for the differences between the samples.
Line 357: phenylpropyl aldehyde: 3-phenylpropanal
Line 359: Please delete neral. The percentage of neral in the SDE extract of sample A was higher.
3.4. Quantitation of Important odorants and Calculation of OAV
I think that application of the OAV experiment does not provide the potent activity of unknown compounds. Therefore, in this study, Unknown 1 (almond, caramel, toasty), the most potent odorant in the AEDA experiment, was not determined in the OAV experiments. The authors should mention it in the manuscript.
Lines 368 and 371: in the extracts of two types of braised pork
Table 5
LOD and LOQ: μg/g?
Lines 376-379: 3-Methylbutanal and Furfuryl alcohol are missing.
Line 377: phenyl acetaldehyde: phenylacetaldehyde
Line 377: lauric aldehyde: dodecanal
4.Conclusions
Line 396: One hundred nine
Line 397: “Combined with the GC-MS and GC-O”? The result was obtained from the OAV experiment. The result by AEDA should be mentioned.
Lines 400-402: The conclusion is not correct because the volatile compounds are rich in the SDE extract of sample B.
Author Response
Point 1: Abstract
Two types of braised pork were prepared from self-made braised sauce added to Maillard reaction intermediate (MRI) and white granulated sugar, respectively. Descriptive sensory analysis and gas chromatography-mass spectrometry (GC-MS) were conducted to investigate their differences in sensory and aroma compounds. The results showed that the effect of self-made braised sauce in braised pork was comparable to the white granulated sugar. One hundred nine volatile flavor compounds were identified by GC-MS using headspace-solid phase microextraction (HS-SPME) and simultaneous distillation and extraction (SDE). Thirty-six odor active compounds with retention index ranged from 935–2465 were identified by aroma extract dilution analysis (AEDA). Additionally, their odor activity values (OAV) were calculated. It was found that 17 aroma compounds showed an OAV greater than 1. Among them, pentanal (almond, pungent), nonanal (fat, green), ((E, E)-2,4-decadienal (fat, roast), phenylacetaldehyde (hawthorn, honey, sweet), dodecanal (lily, fat, citrus) and linalool (floral, lavender) reached the highest OAV values (>200), indicating a significant contribution to the aroma of two types of braised pork. These results indicated that the self-made braised sauce added with MRI could be used for cooking braised pork with good sensory characteristics.
Response: According to suggestion of reviewer, we have revised the abstract.
Point 2:1.Introduction
Line 50: xylose-l-cysteine
Response: The change has been made, as can be seen in Page 2, line 54.
Point 2: Line 51: gas chromatography-olfactometry
Response: The change has been made, as can be seen in Page 2, line 55.
Point 3: 2.Materials and Methods
Line 68: 3-Methylbutanal
Response: The change has been made, as can be seen in Page 2, line 73.
Point 4: Line 70: phenylacetaldehyde
Response: The change has been made, as can be seen in Page 2, line 75.
Point 5: Lines 70 and 72: Were “(E)-cinnamaldehyde” and “cinnamaldehyde” different compounds?
Response: They are all cinnamaldehyde, but they are isomers.
Point 6: Line 71: 3-phenylpropanal
Response: The change has been made, as can be seen in Page 2, line 76;Page 13, line 340; Page 16, line 388; table 4 and 5.
Point 7: Line 72: 4-Methoxybenzaldehyde
Response: It is indeed 2-Methoxybenzaldehyde.
Point 8: Line 73: Please delete “. All of the standard chemicals above”. “…and coumarin were purchased from …”
Response: The change has been made, as can be seen in Page 2, line 79-80.
Point 9: Line 74: Please delete “and were of GC quality”.
Response: The change has been made, as can be seen in Page 2, line 80.
Point 10: Line 78: Please replace “in n-alkanes” by “in hexane”.
Response: The change has been made, as can be seen in Page 2, line 84.
Point 11: Line 81: Please describe the location of Anhui Muyang Oil and Fats Co., Ltd.
Response: The location of Anhui Muyang Oil and Fats Co., Ltd. has been added, as can be seen in Page 2, line 87.
Point 12: Line 82: 2.3. Preparation of self-made braised sauce
Response: The change has been made, as can be seen in Page 2, line 88.
Point 13: Lines 83-84: Xylose (4.5 g) and l-cysteine (0.54 g)(molar ratio about 10:1.5) were dissolved in water (50 g).
Response: The change has been made, as can be seen in Page 2, line 90-91.
Point 14: Lines 82-94 and 94-106: This preparation should be carried out at least in triplicate because the recipe is extremely complex.
Response: We agree with the reviewer’s comment. In fact, we repeated them at least three times during the experiment. And the sentences have been added in revised manuscript.
Point 15: Line 113: In Results, it is better to show the scores obtained (means and standard deviation) in a table.
Response: According to suggestion of reviewer, the sensory evaluation score results have been added as in table and figure.
Point 16: Line 125: “caramel” flavor
Response: The change has been made, as can be seen in Page 3, line 127.
Point 17: Lines 125-126: In Results, the scores obtained (means and standard deviation) should be shown in a table.
Response: According to suggestion of reviewer, the sensory evaluation score results have been added as in table.
Point 18: Lines 133-134: Please delete “at 250°C for 5 min”.
Response: It has been deleted, as can be seen in Page 3, line 142-143.
Point 19: Line 135-145: The authors should carry out the SAFE procedure instead of the SDE procedure.
Response: The common methods adopted in the pretreatment of samples for the analysis of the volatile components include organic solvent extraction, solvent assisted flavor evaporation (SAFE), simultaneous distillation extraction (SDE), solid-phase microextraction (SPME), supercritical CO2 fluid extraction, etc. At present, SAFE, SDE and HS-SPME can be widely used in the distillation of the volatile components from the complex matrix materials. Among of them, SAFE is a mild treatment isolating both volatile and semi-volatile compounds without matrix influence. So it is very popular now. However, different food systems may require different extraction and enrichment methods, but it takes compare analysis to get a conclusion.
Inspired by the reviewer, we can study the effects of three extraction methods (SAFE, SPME and SDE) on braised pork flavor in future. Meanwhile, we hope the reviewer can give us this opportunity.
Point 20: Line 139: under nitrogen atmosphere?
Response: Not in nitrogen atmosphere but in the normal atmosphere.
Point 21: Line 140: collect in dichloromethane
Response: The change has been made, as can be seen in Page 4, line 150.
Point 22: Line 141: Na2SO4
Response: The change has been made, as can be seen in Page 3, line 151.
Point 23: Line 144: The weight (g) of the extracts obtained should be shown for the quantitation
Response: The concentration is calculated by the braised pork sample (μg/g braised pork), so the weight (g) of the extract obtained was not weighed.
Point 24: Line 163: The split ratio was 10:1. The eluate was split…
Response: The change has been made, as can be seen in Page 4, line 173.
Point 25: Line 183: Zero point zero one
Response: The change has been made, as can be seen in Page 4, line 193.
Point 26: Line 185: Were the ratios 1:10 (1/11)-1:60 (1/61) equal to 10-fold-60-fold dilutions?
Response: They mean 10-fold-60-fold dilutions.
Point 27: Lines 191-194: Were the concentrations the values in extract or braised pork?
Response: The concentrations were the values in braised pork.
Point 28: Line 198: detection threshold concentration in water (μg/g)
Response: The change has been made, as can be seen in Page 5, line 208.
Point 29: 3. Results
3.1. Sensory Evaluation of the Braised Pork Samples
As mentioned above, the authors should show the sores (means and standard deviation) in a table. If possible, please also show the spider charts in a figure. There is no need for Figure 1 and the table on F-values.
Response: According to suggestion of reviewer, the sensory evaluation score results have been added as in table 1 and spider charts in figure 1.
Point 30: Lines 231-233: The reason is not correct because the volatile compounds, such as furfural, 5-methylfurfural, 2,3-dihydro-3,5-dihydroxy-6-methyl-4H-pyran-4-one etc., were generated in sample A in larger amounts than in sample B. These compounds might be generated via caramelization of sugars.
Response: We agree with the reviewer’s comment. The reason has been revised, as can be seen in Page 7, line 248-252.
Point 31: 3.2 Volatile Flavor Compounds Identified from the Braised Pork
Lines 243-244: The authors mentioned the advantages of the SPME method. Among these, there is the advantage such as “the volatile compounds detected do not change”. The authors should have been paying attention to the change of volatile compounds. Therefore, the authors should not apply the SDE method.
Response: There are several methods to analysis the flavor. However, each method has its own advantages and disadvantages. In this study, the two methods can be deemed as a complement for each other during the volatility analysis of braised pork. In contrast, taking into account the quantification, the SDE is more accurate although the SPME extracted more hydrocarbons, sulphur and nitrogen compounds.
Point 32: Lines 248-249: “All 109 volatile…significantly different (p <0.05)”. The authors should show the evidences. The authors should confirm the correlations between sensory attributes and volatile flavor compounds by multivariate analysis.
Response: Because the contents of volatile compounds are relative in this section. So we did not do the correlations between sensory attributes and volatile flavor compounds by multivariate analysis. We have deleted the statement, as can be seen in Page 7, line 267-268.
Point 33: Line 249: Seventy-seven
Response: The change has been made, as can be seen in Page 7, line 269.
Point 34: Line 252: “The difference…principle and conditions”. This reason is too common.
Response: The change has been made, as can be seen in Page 7, line 272-273.
Point 35: Lines 257-259: The authors should not apply the SDE method. The sulfur-containing compounds such as furfuryl mercaptan and 2-methyl-3-furanthiol may be contributors to the overall aroma of meat samples.
Response: We agree with the reviewer’s comment about sulfur-containing compounds such as furfuryl mercaptan and 2-methyl-3-furanthiol may be contributors to the overall aroma of meat samples. In this study, we focused on the overall profile of the sample and the accurate quantification of the compounds. So, we applied the SDE method. Actually, we also considered the importance of sulfur-containing compounds, and we will make an in-depth analysis of sulfur-containing compounds in combination with GC-FPD in future studies.
Point 36: Line 260: “that might lead to…some volatile compounds”. The authors should apply the SAFE method instead of the SDE method.
Response: During the SDE process, some complex side reactions and the loss of some volatile compounds do occur. In this study, we focused on the overall profile of the sample and the accurate quantification of the compounds. So, we applied the SDE method. In addition, we can study the effects of three extraction methods (SAFE, SPME and SDE) on braised pork flavor in future.
Point 37: Table 2
o-Cymene: p-Cymene?
2-Methyl-propanal: 2-Methylpropanal
3-Methyl-butanal: 3-Methylbutanal
2-Methyl-butanal: 2-Methylbutanal
Phenyl acetaldehyde: Phenylacetaldehyde
2,4-Decadienal: (E,E)-2,4-Decadienal
Anisic aldehyde: 4-Methoxybenzaldehyde
Ethyl digol: 2-(2-Ethoxyethoxy)ethanol
Diacetone alcohol: 4-Hydroxy-4-methyl-2-pentanone
Methylisohexenyl ketone: 6-Methyl-5-hepten-2-one
Methyl butyrate: Methyl butanoate
γ-Butyrolactone: category of “Lactone”
Estragol: Estragole
2-Propene-1-thiol: Allyl mercaptan
Methyl allyl sulfide: Allyl methyl sulfide
Methylthiophene: 2 or 3-Methylthiophene?
Allyl sulfide: Diallyl sulfide
2-Methyl-3-sulfanylfuran: 2-Methyl-3-furanthiol
2-Acetyl thiazole: 2-Acetylthiazole
5-Methylfurfural: 5-Methyl-2-furfural
5-Hydroxymethylfurfural: 5-Hydroxymethyl-2-furfural
Response: According to suggestion of reviewer, the changes have been made. Among them, o-Cymene and p-Cymene are different, the former is ortho and the latter is opposite.
Point 38: Table 3
o-Cymene: p-Cymene?
3-Methyl- butanal: 3-Methylbutanal
(E)-Cinnamaldehyde and Cinnamaldehyde: different compounds?
Lauric aldehyde: Dodecanal
Anisic aldehyde: 4-Methoxybenzaldehyde
E-2-Tetradecen-1-ol: (E)-2-Tetradecen-1-ol
Heptanone: 2, 3, or 4-Heptanone ?
Hydroxyacetone: Acetol
γ-Nonanolactone: γ-Nonalactone
Ethyl cinnamate: category of “Ester”
2,6-Di-tert-butyl-4-methylphenol and 2,4-Di-tert-butylphenol: These compounds are food additives as antioxidant. Please delete these compounds.
Estragol: Estragole
5-Methylfurfural: 5-Methyl-2-furfural
Response: According to suggestion of reviewer, the changes have been made. (E)-Cinnamaldehyde and Cinnamaldehyde are different compounds, they are isomers.
Point 39: line281: Are “the semi-quantitation results” GC peak area percentages?
Response: In this section, the semi-quantification results mean GC peak area percentages.
Point 40: Lines 283-287: These sentences should be deleted because of the discussion according to the later AEDA and OAV experiments.
Response: According to suggestion of reviewer, the changes have been made.
Point 41: Line 288: 3-methyl-butanal: 3-methylbutanal
Response: The change has been made, as can be seen in Page 12, line 310.
Point 42: Line 289: phenyl acetaldehyde: phenylacetaldehyde
Response: The change has been made, as can be seen in Page 12, line 311.
Point 43: Line 289: 2,4-decadienal: (E,E)-2,4-decadienal
Response: The change has been made, as can be seen in Page 12, line 311.
Point 44: Line 289: anisic aldehyde: 4-methoxybenzaldehyde
Response: The change has been made, as can be seen in Page 13, line 312.
Point 45: Line 290: cinnamaldehyde: (E)-cinnamaldehyde or cinnamaldehyde?
Response: In this section, cinnamaldehyde means the cis- cinnamaldehyde.
Point 46: Lines 296-298: This reason is not correct because the major sulfur compounds are allyl mercaptan derivatives which are not generated via Maillard reaction in meat but via degradation of the precursors in garlic material.
Response: We agree with the reviewer’s comment. We have revised the explanation, as can be seen in Page 12, line 320-321.
Point 47: Line 301: This reason is incompatible with the sentence described at lines 243-244. Further, the SPME procedure is carried out in a sealing vial. Therefore, the loss of volatiles might be easily occurred.
Response: We agree with the reviewer’s comment. The loss of volatiles might be easily occurred in SPME procedure. Methanthiol was highly volatile, so it is easy to adsorb in the fiber and loss in SDE procedure.
Point 48: Line 302: Was leucine derived from the pork meat?
Response: Leucine is an amino acid found in meat. We also measured free amino acids and total amino acids in pork. The results showed that leucine is found in meat.
Point 49: Line 303: Please delete “and methionine”.
Response: “and methionine” has been deleted, as can be seen in Page 13, line 327.
Point 50: Line 303: “It can provide… fruity odor characteristics”. This is not correct. Please delete.
Response: It has been deleted, as can be seen in Page 13, line 327-328.
Point 51: Lines 304-305: 3-Methylbutanal may not contribute to the caramel odor. This sentence is not correct. Please delete.
Response: It has been deleted, as can be seen in Page 13, line 327-328.
Point 52: Lines 308-309: Please delete the sentences because the thiophene compounds were not important odorants.
Response: It has been deleted.
Point 53: Line 316: phenyl acetaldehyde: phenylacetaldehyde
Response: The change has been made, as can be seen in Page 14, line 340.
Point 54: Line 316: benzenepropanal: 3-phenylpropanal
Response: The change has been made, as can be seen in Page 14, line 340.
Point 55: Line 317: only one lactone γ-undecalactone was detected
Response: The change has been made, as can be seen in Page 14, line 341.
Point 56: Lines 317-319: This is not correct. Please delete.
Response: The change has been made, as can be seen in Page 14, line 342-344.
Point 57: 3.3. Identification of the Odor-Active Compounds in Braised Pork
Line 321: in braised pork: in the extracts of two types of braised pork
Response: The change has been made, as can be seen in Page 14, line 349.
Point 58: Lines 321-322: Please delete “by SDE-AEDA-GC-MS/O”. Please replace by “by AEDA”.
Response: The change has been made, as can be seen in Page 14, line 350.
Point 59: Line 323: 1 ketone
Response: The change has been made, as can be seen in Page 14, line 351.
Point 60: Line 323: 1 phenol
Response: The change has been made, as can be seen in Page 14, line 352.
Point 61: Line 323-324: 1 nitrogenous compound
Response: The change has been made, as can be seen in Page 14, line 352.
Point 62: Line 324: Thirty
Response: The change has been made, as can be seen in Page 14, line 353.
Point 63: Line 327: RI 1135-2053
Response: The change has been made, as can be seen in Page 14, line 355.
Point 64: Line 327: Furfuryl alcohol does not have a “meaty” odor.
Response: It has been deleted, as can be seen in Page 14, line 356.
Point 65: Line 329-331: The authors mentioned the potent odorants in “braised pork”. However, the AEDA results are limited to the SDE extracts. Therefore, this sentence is not correct.
Response: We have revised the statement, as can be seen in Page 14, line 356.
Point 66: Line 332: Table 4 Odor-active compounds detected in the SDE extracts of two types of braised pork by AEDA
Response: The change has been made.
Point 67: Table 4
I think that maltol, furaneol, homofuraneol, sotolone, and abhexone, which are Maillard reaction (or caramelization) products, may be the contributors because of having a low odor threshold. However, these compounds were not determined by GC-O and AEDA in this study. The reason is the application of the SDE method.
Response: We agree with the reviewer’s comment. These compounds were not detected in SDE method precisely because of the method's shortcomings. Thus, in our further study, we should combine a variety of methods for flavor identification including SAFE method.
Point 68: (E)-Cinnamaldehyde and Cinnamaldehyde: different compounds?
Response: They are isomers.
Point 69: Phenyl acetaldehyde: Phenylacetaldehyde
Lauric aldehyde: Dodecanal
Anisic aldehyde: 4-Methoxybenzaldehyde
γ-Undecalactone: category of “Lactone”
Response: The change has been made.
Point 70: In the column of Odor Threshold and Line 340: Were all values measured? Why can you measure the values of (E)-cinnamaldehyde, 2-methoxybenzaldehyde, and 2-methyltetrahydrofuran-3-one? These standard compounds can be obtained from the suppliers.
Response: The odor thresholds of all these compounds were obtained by consulting the literatures. And these standard compounds can be obtained from the suppliers (Aladdin biochemical technology co., LTD (Shanghai, China)).
Point 71: Line 336: Do the concentrations indicate the value in extracts or in pork?
Response: The concentrations indicate the value in pork.
Point 72: Line 341 and 342: Please replace “cooked by” by “with”.
Response: The change has been made, as can be seen in Page 15, line 371-372.
Point 73: Lines 343-346: Please delete.
Response: The change has been made.
Point 74: Lines 352-353: The addition of spices is a common operation for sample A and B. The sentence is not reason for the differences between the samples.
Response: The sentence has been deleted, as can be seen in Page 15, line 382-383.
Point 75: Line 357: phenylpropyl aldehyde: 3-phenylpropanal
Response: The change has been made, as can be seen in Page 16, line 388.
Point 76: Line 359: Please delete neral. The percentage of neral in the SDE extract of sample A was higher.
Response: It has been deleted, as can be seen in Page 16, line 389.
Point 77: 3.4. Quantitation of Important odorants and Calculation of OAV
I think that application of the OAV experiment does not provide the potent activity of unknown compounds. Therefore, in this study, Unknown 1 (almond, caramel, toasty), the most potent odorant in the AEDA experiment, was not determined in the OAV experiments. The authors should mention it in the manuscript.
Response: We agree with the reviewer’s comment. Due to our carelessness, the concentrations of unknown compounds were not calculated. And the relevant interpretation has been added in revised manuscript.
Point 78: Lines 368 and 371: in the extracts of two types of braised pork
Response: The change has been made.
Point 79: Table 5 LOD and LOQ: μg/g?
Response: We have checked the data and the unit is μg/Kg.
Point 80: Lines 376-379: 3-Methylbutanal and Furfuryl alcohol are missing.
Response: 3-Methylbutanal has been added, Furfuryl alcohol with OAV<1.
Point 81: Line 377: phenyl acetaldehyde: phenylacetaldehyde
Response: The change has been made.
Point 82: Line 377: lauric aldehyde: dodecanal
Response: The change has been made.
Point 83: 4.Conclusions
Line 396: One hundred nine
Response: The change has been made, as can be seen in Page 17, line 430.
Point 84: Line 397: “Combined with the GC-MS and GC-O”? The result was obtained from the OAV experiment. The result by AEDA should be mentioned.
Response: The change has been made, as can be seen in Page 17, line 431-432.
Point 85: Lines 400-402: The conclusion is not correct because the volatile compounds are rich in the SDE extract of sample B.
Response: The statement has been changed.
Reviewer 2 Report
The authors had investigated the sensory characteristics and volatile profile of braised pork. I applaud the authors for designing the study but additional analysis and a discussion section is needed for this paper.
Section 2.
Consider changing 2.2 materials to food sample
Section 2.5
Line 114. Incomplete (21 ± ???°C)
The sensory was carried out first on desirability on 1-10 scale and after a DA was carried out on 9 point scale (0-9). The authors needs to justify why this was done as usually a 9-point scaling is desired for both measures, unless if the authors are trying to measure hedonic (liking) for the first measurement.
Results
Fig 1. Having the F value as figure doesn't add value, please merge with Table 1.
Table 1. It would also be worth instead of putting the F value in the table try putting the mean intensity rating value instead so that the readers can see which is sig. higher to which as such similarly to Table 2
Line 219, clearly an interaction was investigated here, please amend the data analysis section that a 2 way ANOVA was carried out?
Table 2 and 3, looking at the vast amount of the data collected here it seems that the authors looked at the effect of SPME vs SDE. Perhaps the authors should consider attempting an additional analysis to include method as a factor and multivariate analysis to further infer these results as it is very difficult to see what is happening.
Discussion seems lacking/missing. Based on line 59-65 the authors had stated several research questions and this needs to be answered, whether their results are supported with the current lit. or not.
Author Response
Point 1: Section 2.
Consider changing 2.2 materials to food sample
Response: The change has been made, as can be seen in Page 2, line 81.
Point 2: Section 2.5
Line 114. Incomplete (21 ± ???°C)
The sensory was carried out first on desirability on 1-10 scale and after a DA was carried out on 9 point scale (0-9). The authors needs to justify why this was done as usually a 9-point scaling is desired for both measures, unless if the authors are trying to measure hedonic (liking) for the first measurement.
Response: According to the comment of reviewer 3, this sentence has been deleted. There were no direct links between the two groups of sensory evaluations. The 7 aroma attributes including meat flavor, spicy flavor, greasy, sauce flavor, braised flavor, caramel flavor and overall flavor were the detail attribute evaluation of aroma.
Point 3: Results
Fig 1. Having the F value as figure doesn't add value, please merge with Table 1.
Response: According to the suggestions of reviewer 1, Figure 1 has been deleted. And we added the radar Figure of sensory evaluation.
Point 4: Table 1. It would also be worth instead of putting the F value in the table try putting the mean intensity rating value instead so that the readers can see which is sig. higher to which as such similarly to Table 2
Response: According to the suggestion of reviewer, we have added the mean intensity of aroma attributes, as can be seen in new table 1.
Point 5: Line 219, clearly an interaction was investigated here, please amend the data analysis section that a 2 way ANOVA was carried out?
Response: All data have been carried out by 2 way ANOVA.
Point 6: Table 2 and 3, looking at the vast amount of the data collected here it seems that the authors looked at the effect of SPME vs SDE. Perhaps the authors should consider attempting an additional analysis to include method as a factor and multivariate analysis to further infer these results as it is very difficult to see what is happening.
Response: We are obliged to you for your advice. We attempt to use other methods for further comparative analysis. However, since the data obtained by the two methods were not been quantified, statistical comparison cannot be made.
Point 7: Discussion seems lacking/missing. Based on line 59-65 the authors had stated several research questions and this needs to be answered, whether their results are supported with the current lit. or not.
Response: Because our ability is limited, the results and discussion sections may not be comprehensive. But we think the results have elucidated the current research questions.

Reviewer 3 Report
The study is an interesting work trying to describe the use of a man-made substitute for sugar in traditional braised pork. Such work, although specifically described for Chinese braised pork has application to other products around the world that use a sauce made from sugar and other components to give flavors associated with traditional Maillard browning and carmelization.
What is not clear is the real benefit to the consumer of doing this. The authors provide no information on the calorie reduction that would occur from this. According to my calculation it would be less than 50 calories per batch of braised pork they made using 500g of pork. If a typical serving is 1/3 to 1/2 of that pork with sauce, the calories saved is minimal - less than 20. This seems like a lot of trouble for so few calories and also means that the ingredients are now "artificial" as opposed to natural. Does this matter? The authors do not mention any issues with this, but consumers in most parts of the world would wonder about this. More discussion is needed. I am not suggesting that this means the work is not valuable, but that the authors overlook some important information in their rush to provide a rationale for the research. I think there is some valuable information here, but it goes beyond simply making a new sauce and really is about the ability to "recreate" Maillard flavors without sugar (although with the use of sugar alcohols). That is what should be discussed more both in the introduction and the results.
The "accepance" data (1-9 scale of desirability) must be removed from the paper and any reference to it needs to be removed from the results and discussion. Foods follows the Recommendations for Publications Contaoiniong Sensory Data (see the author instructions https://www.mdpi.com/journal/foods/instructions and scroll down near the bottom. Those guidelines, published by the Society of Sensory Professionals clearly state that acceptability data should never be collected on a small "trained" or oriented panel. It takes far more people to measure liking.
The descriptive sensory data (0-9 scale of intensity) is OK and can remain. It would help if you could describe the aroma notes more specifically. The notes you list generally are rather vague. I understand the use of them in this article, but I don't really know what they mean. For example "sauce" aroma could mean a lot of things. You give a reference of a "type of braise soy sauce" to represent sauce note, but does that mean the "sauce" note smells like soy sauce. I don't think that is what you intend to say, but I don't know what the reference is. This is the same for other notes. More description would help.
Figure 1 provide no information about the samples. Much more valuable would be a bar graph with error bars that shows the mean value of each attribute for each product. The product scores placed sis-by-side would make it clear which product has the higher score and by how much. It would be easy to state in the text results and in the figure that th two samples were different for all attributes at a P value of < or = to 0,05.
Similarly, I don't know what Table 1 really shows that couldn't be described very briefly in the paper without taking up space for the table. You did a 3 way ANOVA and only product gave a significant difference. Panelist and replication (which I assume was a remake of the product and not just a re-sampling of the same product by the panelists - this needs to be clear in the analysis) were not different and neither were any interactions, except one interaction for sample by replication for overall flavor (data not shown).
I wish that you would separate the first section of the volatile compound data into two sections for discussion. First, discuss the differences in the two methods and your comments that they provide complementary data and both should be used. Then, second, talk about how the compounds relate to flavor, etc. The two discussions are merged together in the first couple of paragraphs and it makes the two different objectives less easy to follow.
Tables 2 and 3 need to be single spaced. You also might highlight (with colored text) those compounds in each method that were NOT found in the other method to show quickly how the two methods differ in the compounds they find - a key finding of your study.
Also, have these methods never been compared before? If they have you should have specific references and discussion about your results vs those.
I wish that you would mention the fact that using OAV does not imply a specific intensity of the sensory properties of the product because the slope of the curve for intensity above threshold varies greatly for different compounds. You don't really say that it does, but it might help people understand that you really are looking at general trends of potential odors when using OAV.
Please do not use the word "significant" when you mean important. Significant should be reserved for statistical comparisons.
A native English scientific writer needs to thoroughly review the paper for grammar and sentence structure. It is not bad, but definitely needs to be improved.
Author Response
Point 1: What is not clear is the real benefit to the consumer of doing this. The authors provide no information on the calorie reduction that would occur from this. According to my calculation it would be less than 50 calories per batch of braised pork they made using 500g of pork. If a typical serving is 1/3 to 1/2 of that pork with sauce, the calories saved is minimal - less than 20. This seems like a lot of trouble for so few calories and also means that the ingredients are now "artificial" as opposed to natural. Does this matter? The authors do not mention any issues with this, but consumers in most parts of the world would wonder about this. More discussion is needed. I am not suggesting that this means the work is not valuable, but that the authors overlook some important information in their rush to provide a rationale for the research. I think there is some valuable information here, but it goes beyond simply making a new sauce and really is about the ability to "recreate" Maillard flavors without sugar (although with the use of sugar alcohols). That is what should be discussed more both in the introduction and the results.
Response: We agree with the reviewer’s comment. I also think it is very meaningful on the calorie reduction. In our study, we first want to know if the flavor of braised pork with less sugar sauce can achieve the flavor of traditional braised pork. So, we just only compared the differences in flavor in this study. In the future, we are going to continue to study how many calories are being reduced when sugar goes down, and we will also study the substitution of natural products for artificial in the sauce preparation.
Point 2: The "accepance" data (1-9 scale of desirability) must be removed from the paper and any reference to it needs to be removed from the results and discussion. Foods follows the Recommendations for Publications Contaoiniong Sensory Data (see the author instructions https://www.mdpi.com/journal/foods/instructions and scroll down near the bottom. Those guidelines, published by the Society of Sensory Professionals clearly state that acceptability data should never be collected on a small "trained" or oriented panel. It takes far more people to measure liking.
Response: According to the suggestion of reviewer, we have removed them from the paper.
Point 3: The descriptive sensory data (0-9 scale of intensity) is OK and can remain. It would help if you could describe the aroma notes more specifically. The notes you list generally are rather vague. I understand the use of them in this article, but I don't really know what they mean. For example "sauce" aroma could mean a lot of things. You give a reference of a "type of braise soy sauce" to represent sauce note, but does that mean the "sauce" note smells like soy sauce. I don't think that is what you intend to say, but I don't know what the reference is. This is the same for other notes. More description would help.
Response: In order for the panelists to have a consistent feeling about these 7 aroma attributes, we just only choose a more similar smell as a reference. Also, we have read a lot of references, and they used the same method.
Point 4: Figure 1 provide no information about the samples. Much more valuable would be a bar graph with error bars that shows the mean value of each attribute for each product. The product scores placed sis-by-side would make it clear which product has the higher score and by how much. It would be easy to state in the text results and in the figure that th two samples were different for all attributes at a P value of < or = to 0,05.
Response: According to the suggestion of reviewer, we have changed the figure 1 and table 1 with new figure 1 and table 1 with score values.
Point 5: Similarly, I don't know what Table 1 really shows that couldn't be described very briefly in the paper without taking up space for the table. You did a 3 way ANOVA and only product gave a significant difference. Panelist and replication (which I assume was a remake of the product and not just a re-sampling of the same product by the panelists - this needs to be clear in the analysis) were not different and neither were any interactions, except one interaction for sample by replication for overall flavor (data not shown).
Response: We have deleted the original Table 1.
Point 6: I wish that you would separate the first section of the volatile compound data into two sections for discussion. First, discuss the differences in the two methods and your comments that they provide complementary data and both should be used. Then, second, talk about how the compounds relate to flavor, etc. The two discussions are merged together in the first couple of paragraphs and it makes the two different objectives less easy to follow.
Response: According to the suggestion of reviewer, we have separated the first section of the volatile compound data into two sections for discussion.
Point 7: Tables 2 and 3 need to be single spaced. You also might highlight (with colored text) those compounds in each method that were NOT found in the other method to show quickly how the two methods differ in the compounds they find - a key finding of your study.
Response: According to the suggestion of reviewer, we have changed the line spacing. And we have highlighted those compounds in each method that were not found in the other method.
Point 8: Also, have these methods never been compared before? If they have you should have specific references and discussion about your results vs those.
Response: Several articles have been published comparing the two methods, but they are all for different objects. Such as “Chinese Sinkiang fermented camel milk” (Food Chemistry, 129 (2011), 1242–1252), sparkling wines (Food Chemistry, 105(2007), 428–435) and so on. It was hard to find the same compounds, so we did not cite references.
Point 9: I wish that you would mention the fact that using OAV does not imply a specific intensity of the sensory properties of the product because the slope of the curve for intensity above threshold varies greatly for different compounds. You don't really say that it does, but it might help people understand that you really are looking at general trends of potential odors when using OAV.
Response: We agree with the reviewer’s comment. We have revised some statements in revised manuscript.
Point 10: Please do not use the word "significant" when you mean important. Significant should be reserved for statistical comparisons.
Response: According to the suggestion of reviewer, we have changed the sentences.
Point 11: A native English scientific writer needs to thoroughly review the paper for grammar and sentence structure. It is not bad, but definitely needs to be improved.
Response: We have asked a native English scientific writer to help us revise the manuscript. So we think it should read more smoothly now.

Round 2
Reviewer 1 Report
Point 45: Line 290: cinnamaldehyde: (E)-cinnamaldehyde or cinnamaldehyde?
Response: In this section, cinnamaldehyde means the cis- cinnamaldehyde.
Please replace all of “cinnamaldehyde” by “(Z)-cinnamaldehyde” in the manuscript.
Point 64: Line 327: Furfuryl alcohol does not have a “meaty” odor.
Response: It has been deleted, as can be seen in Page 14, line 356.
Please delete the “meaty” of Furfuryl alcohol in Table 4.
Author Response
Point 1: Please replace all of “cinnamaldehyde” by “(Z)-cinnamaldehyde” in the manuscript.
Response: The change has been made, as can be seen in Page 2, line 78; table 1; table 2; Page 12, line 316; Page 13, line 345; table 3; Page 15, line 391, 393; table 4; Page 17, line 427.
Point 2: Please delete the “meaty” of Furfuryl alcohol in Table 4.
Response: The change has been made, as can be seen in Table.

Reviewer 2 Report
I'd like to thank the authors for their responses. However, there are some other adjustments that needs to be done.
Figure 1. It seems the F value bar charts hasn't been deleted yet. The spider plot looks good, but consider replacing "sample A" and "sample B" into something more informative.
Table 1 isn't really needed anymore as the spider plot is more than sufficient.
Section 2.14 The authors had mentioned that they have carried out 2 way ANOVA, please amend this section as it still states one-way ANOVA.
Re multivariate, the experiment had produced peak area% for both samples which seemed to be a form of quantification? Perhaps a PCA map for sample A/B with SPME and SDE as sample loading would provide a good snapshot.
Re Point 7. Having a limited ability is not a good way on addressing comments. I'm asking the authors to add more reflection here on their results a quick compare and contrast on what is out there in the literature to further validate their results whether it is in agreement or in disagreement. For example, the authors had attempted to explain the formation of the compounds (Section 3.2) this approach should be applied for the other sections in the discussion.
Author Response
Point 1: Figure 1. It seems the F value bar charts hasn't been deleted yet. The spider plot looks good, but consider replacing "sample A" and "sample B" into something more informative.
Response: The F value bar chart has been deleted. And the information of "sample A" and "sample B" has been added in Figure 1 caption.
Point 2: Table 1 isn't really needed anymore as the spider plot is more than sufficient.
Response: We have deleted the Table 1.
Point 3: Section 2.14 The authors had mentioned that they have carried out 2 way ANOVA, please amend this section as it still states one-way ANOVA.
Response: Actually, multiple comparisons were used one-way ANOVA, and interaction analysis was used 3-way ANOVA. And the change has been made, as can be seen in Page 3, line 213-214.
Point 4: Re multivariate, the experiment had produced peak area% for both samples which seemed to be a form of quantification? Perhaps a PCA map for sample A/B with SPME and SDE as sample loading would provide a good snapshot.
Response: We agree with the reviewer’s comment. It would be better if we can do PCA map for sample A/B with SPME and SDE as sample loading would provide a good snapshot. However, peak area% can only be used to compare the relative content of compounds in the same sample, while the contents of compounds in different samples could not be compared because there is no guarantee that all the compounds in the samples have been detected. Therefore, the content calculated by the percentage of peak area is very inaccurate.
Point 5: Re Point 7. Having a limited ability is not a good way on addressing comments. I'm asking the authors to add more reflection here on their results a quick compare and contrast on what is out there in the literature to further validate their results whether it is in agreement or in disagreement. For example, the authors had attempted to explain the formation of the compounds (Section 3.2) this approach should be applied for the other sections in the discussion.
Response: We agree with the reviewer’s comment. We have added some related explanations and references in revised manuscript, as can be seen in Page 16, line 396-405; Page 19, line 532-537.

Reviewer 3 Report
Thank you for making many of the changes suggested. There are a few changes that I would still recommend be made
Line 118-119 starting "They were trained for 2 weeks..." should be left in and the following paragraph merged with this paragraph. Although it was stated in relation to the acceptance data, I am sure that contributed to the training for the descriptive portion as well. Only the sentence that starts "the color..." should be removed since that relates to the acceptance portion ( note the last sentence has already been moved to the next paragraph.
Although
you state that other authors refer to the lexicon and references as you
do, they also frequently include a "brand" or specific information
about that reference. For example, see
Wang, Hongwei ; Zhang, Xiao ; Suo, Huayi ; Zhao, Xin ; Kan, Jianquan. 2019. Aroma and flavor characteristics of commercial Chinese traditional bacon from different geographical regions. Journal of Sensory Studies, 2019, Vol.34: e12475
from another group of Chinese Scientists
and these two articles from recent issues of this journal, Foods (the first an article evaluating a food product and the second a review article on using lexicons).
Lipan, Leontina; Cano-Lamadrid, Marina; Corell, Mireia; Sendra, Esther; Hernández, Francisca; Stan, Laura; Vodnar, C. Dan; Vázquez-Araújo, Laura; Carbonell-Barrachina, A. Ángel. 2019. Sensory Profile and Acceptability of HydroSOStainable Almonds. Foods: 8, 64
Suwonsichon, Suntaree. 2019. The Importance of Sensory Lexicons for Research and Development of Food Products. Foods, 2019, 8: 27
Line 193: Instead of "Zero point..." the amount should read "Ten mg", which is much easier to understand.
Lines 218-221 These results refer to data from the acceptance test, not the descriptive test and should be removed. You did not measure tenderness and succulence in the descriptive test.
Lines 226-227: Starting with Analysis or Variance ... and ending with ...Figure 1. has already been stated in the methods and does not need to be repeated here - eliminate. Simply start with Significant... (the 3rd sentence) and refer to the Figure and Table there.
There are still some minor grammatical issues, but those can be addressed with the editorial staff, I assume.
Author Response
Point 1: Line 118-119 starting "They were trained for 2 weeks..." should be left in and the following paragraph merged with this paragraph. Although it was stated in relation to the acceptance data, I am sure that contributed to the training for the descriptive portion as well. Only the sentence that starts "the color..." should be removed since that relates to the acceptance portion (note the last sentence has already been moved to the next paragraph.
Response: According to the suggestion of reviewer, the change has been made, as can be seen in Page 3, line 118-125.
Point 2: Although you state that other authors refer to the lexicon and references as you do, they also frequently include a "brand" or specific information about that reference. For example, see
Wang, Hongwei ; Zhang, Xiao ; Suo, Huayi ; Zhao, Xin ; Kan, Jianquan. 2019. Aroma and flavor characteristics of commercial Chinese traditional bacon from different geographical regions. Journal of Sensory Studies, 2019, Vol.34: e12475
from another group of Chinese Scientists
and these two articles from recent issues of this journal, Foods (the first an article evaluating a food product and the second a review article on using lexicons).
Lipan, Leontina; Cano-Lamadrid, Marina; Corell, Mireia; Sendra, Esther; Hernández, Francisca; Stan, Laura; Vodnar, C. Dan; Vázquez-Araújo, Laura; Carbonell-Barrachina, A. Ángel. 2019. Sensory Profile and Acceptability of HydroSOStainable Almonds. Foods: 8, 64
Suwonsichon, Suntaree. 2019. The Importance of Sensory Lexicons for Research and Development of Food Products. Foods, 2019, 8: 27
Response: According to the suggestion of reviewer. We have added some related explanations and references in revised manuscript, as can be seen in Page 16, line 396-405; Page 19, line 532-537.
Point 3: Line 193: Instead of "Zero point..." the amount should read "Ten mg", which is much easier to understand.
Response: The change has been made, as can be seen in Page 4, line 194.
Point 4: Lines 218-221 These results refer to data from the acceptance test, not the descriptive test and should be removed. You did not measure tenderness and succulence in the descriptive test.
Response: The change has been made, as can be seen in Page 5, line 218-223.
Point 5: Lines 226-227: Starting with Analysis or Variance ... and ending with ...Figure 1. has already been stated in the methods and does not need to be repeated here - eliminate. Simply start with Significant... (the 3rd sentence) and refer to the Figure and Table there.
Response: The change has been made, as can be seen in Page 6, line 229-231.
Point 6: There are still some minor grammatical issues, but those can be addressed with the editorial staff, I assume.
Response: According to the suggestion of reviewer, we did our best to revise it again.
